# BRIDGE: Triangular Fixed-Point Refinement for Long-Horizon Persona Consistency

Yinghui Jiang [1 2]    Bocheng Xu [2]    Jianye Xie [3 4]    Haotong Sun [2]

## Abstract

Long-horizon dialogue agents suffer from *latent state drift*: what an agent says, what it internally represents, and what it stores in memory can diverge silently across turns. This creates *asymmetric rupture risk*—many locally coherent exchanges undone by a single high-cost contradiction. We propose **BRIDGE** (**B**ehavioral **R**easoning through **I**ntegrated **D**ynamic **G**ated **E**volution), which performs *triangular fixed-point refinement* to explicitly couple Observable ($\mathcal{O}$), Latent ($\mathcal{L}$), and Memory ($\mathcal{M}$) before decoding each response. We prove that under mild conditions, the refinement operator converges to a unique fixed point, providing a theoretical guarantee that the agent's internal state remains self-consistent before each response. Empirically, BRIDGE achieves the highest scores on both PersonaGym (4.59 avg., surpassing Claude-3.7-Sonnet) and CoSER (59.5% avg., +3.1 over Claude-3.7-Sonnet), with gains concentrated on persona-specific metrics (+8.0 Character Fidelity over Qwen2.5-32B-Instruct)—while updating only 0.85% trainable parameters of the frozen backbone. We also provide a Lyapunov-style uniform drift bound for tiered memory updates, grounding bounded persona evolution in long-horizon interaction.

## 1. Introduction

Modern large language models can produce fluent, contextually appropriate responses, yet sustaining persona consistency over long interactions remains unreliable. The characteristic failure mode is not sudden incoherence but gradual *latent state drift*: subtle mismatches between what an agent says, what it implicitly represents, and what it stores in memory accumulate across turns until they surface as contradictions. Over time, even a single contradiction can dominate user perception despite many locally coherent turns, which is particularly consequential for trust-sensitive deployments like mental health support (Park et al., 2023; Li, 2025). Recent benchmarks confirm the severity: even frontier models struggle to maintain persona fidelity under long-horizon interaction (Samuel et al., 2024; Wang et al., 2025a). Notably, scaling does not reliably resolve self-consistency issues (Choi et al., 2024; Frisch & Giulianelli, 2024), suggesting a structural bottleneck: long-horizon interaction lacks an explicit, jointly constrained state representation. Recent work addresses related challenges through Markovian state reconstruction (Chen et al., 2025) and skill-centric state abstractions (Shah et al., 2021), but these approaches lack the triangular coupling and convergence guarantees needed for persona-specific constraints.

Why does drift occur? We trace the root cause to an architectural mismatch: current systems optimize local response quality while treating behavior, reasoning, and memory as loosely coupled components. First, when internal state is not explicitly represented, models generate locally plausible outputs that are weakly anchored to consistent beliefs—an agent may claim to be "introverted" yet repeatedly respond with extroverted enthusiasm (Tseng et al., 2024; Wang et al., 2024). Second, when memory is treated as a flat store, ephemeral details and durable traits are updated without respecting timescale separation, making long-horizon consistency fragile under noise (Zhong et al., 2024; Packer et al., 2023). Third, when adaptation is unconstrained, incremental deviations accumulate into catastrophic drift (Cheng et al., 2024; Wu et al., 2025).

These three factors share a common root: behavior, internal state, and memory are treated as independent modules rather than jointly constrained variables. This motivates two concrete questions:

*(i) Can explicit O–L–M coupling enforce cross-space consistency before decoding?*

[1] National Institute for Data Science in Health and Medicine, Xiamen University, Xiamen, Fujian, China [2] Hangzhou Shenji Technology Co., Ltd, Hangzhou, China [3] College of Computer Science and Technology, China University of Petroleum (East China), China [4] Shandong Key Laboratory of Intelligent Oil and Gas Industrial Software, China. Correspondence to: Haotong Sun <richsun821@gmail.com>.

*Proceedings of the 43rd International Conference on Machine Learning*, Seoul, South Korea. PMLR 306, 2026. Copyright 2026 by the author(s).

*(ii) Can we provide stability controls that prevent unbounded long-horizon drift?*

We answer affirmatively with **BRIDGE** (**B**ehavioral **R**easoning through **I**ntegrated **D**ynamic **G**ated **E**volution), which performs *triangular fixed-point refinement* to couple Observable behaviors ($\mathcal{O}$), Latent states ($\mathcal{L}$), and Memory ($\mathcal{M}$), and pairs this coupling with explicit stability controls. The framework has three components:

- **Triangular fixed-point refinement** (§2.2): A Gauss–Seidel O–L–M refinement operator (Eqs. 1–3) with an explicit sufficient contraction condition guaranteeing a unique fixed point and geometric convergence on a bounded domain (Theorem 2.2).

- **Dual-system control injection** (§2.3): Fast/slow pathways whose outputs are fused (Eq. 8) and injected into backbone hidden states via multiplicative context-aware gating (Eq. 10), providing adaptive computation for *action selection* while triangular refinement enforces cross-space consistency.

- **Hierarchical memory evolution** (§2.4): Three-tier memory with anchored clipped updates (Eq. 11) and a Lyapunov-style uniform drift bound that prevents unbounded identity drift (Theorem 2.3).

BRIDGE achieves the highest average scores on PersonaGym (4.59 PersonaScore) and CoSER (59.5%, +3.1 over Claude-3.7-Sonnet), with particularly large gains on persona-specific metrics (e.g., +8.0 Character Fidelity over Qwen2.5-32B-Instruct). We provide provable guarantees of convergent triangular refinement (Theorem 2.2) and bounded long-horizon drift (Theorem 2.3), further supported by extended runs (Appendix D.1).

## 2. BRIDGE

We introduce **BRIDGE**, which enforces consistency among three explicit agent states—Observable $\mathcal{O}$, Latent $\mathcal{L}$, and Memory $\mathcal{M}$—and injects the resulting control signal through a lightweight hidden-state interface.

Given dialogue history $\mathcal{H}$ and an initial persona $P_0$, the *frozen* LLM backbone encodes the turn-$t$ context into per-token states $\mathbf{H}_t = \{\mathbf{h}_{t,i}\}_{i=1}^n \in \mathbb{R}^{n \times d_h}$. We train only BRIDGE-specific modules under a parameter-efficient fine-tuning setup (Hu et al., 2022; Houlsby et al., 2019), while keeping the backbone weights fixed. BRIDGE maintains a state triple $\mathcal{S}_t = (\mathbf{o}_t, \mathbf{l}_t, \mathbf{m}_t)$ (§2.1) and performs *triangular fixed-point refinement* to obtain $(\mathbf{o}_t^{(K)}, \mathbf{l}_t^{(K)}, \mathbf{m}_t^{(K)})$. We then compose a control signal $\mathbf{C}_t = [\mathbf{o}_t^{(K)}; \mathbf{l}_t^{(K)}; \mathbf{m}_t^{(K)}]$ and inject it via multiplicative context-aware gating to produce conditioned token states $\hat{\mathbf{H}}_t$, from which the standard LM head decodes the response $r_t$.

### 2.1. Problem Formulation

Given a dialogue history $\mathcal{H} = \{(u_1, r_1), \ldots, (u_{t-1}, r_{t-1})\}$ and an initial persona description $P_0$, our goal is to generate $r_t$ that satisfies: (i) *behavioral consistency* with $P_0$, (ii) *cross-space consistency* among internal states, and (iii) *long-horizon stability* against identity drift.

We represent the agent's state at turn $t$ as a tuple $\mathcal{S}_t = (\mathbf{o}_t, \mathbf{l}_t, \mathbf{m}_t)$: $\mathbf{o}_t$ captures observable behaviors, $\mathbf{l}_t$ captures latent cognition, and $\mathbf{m}_t = (\mathbf{m}_t^{(e)}, \mathbf{m}_t^{(a)}, \mathbf{m}_t^{(p)})$ captures hierarchical memory at episodic, affective, and personality timescales. When interfacing with triangular refinement (§2.2), we use its concatenated working form $\tilde{\mathbf{m}}_t = [\mathbf{m}_t^{(e)}; \mathbf{m}_t^{(a)}; \mathbf{m}_t^{(p)}]$. The initial persona description $P_0$ (a text string) is encoded into the initial memory states $\mathbf{m}_0 = (\mathbf{m}_0^{(e)}, \mathbf{m}_0^{(a)}, \mathbf{m}_0^{(p)})$ via a pretrained encoder; these serve as anchors for the bounded memory evolution described in §2.4.

### 2.2. Triangular Fixed-Point Refinement

**Intuition and information flow.** Updating behavior, cognition, and memory independently can introduce silent cross-space mismatches that accumulate over turns. BRIDGE addresses this by refining $(\mathcal{O}, \mathcal{L}, \mathcal{M})$ *jointly* in a closed loop before decoding, using a Gauss–Seidel cycle $\mathcal{M} \to \mathcal{O} \to \mathcal{L} \to \mathcal{M}$ that always consumes the most recent state. We instantiate each edge with *pairwise* cross-attention (rather than a trilinear triangle operator), which keeps the coupling modular and amenable to the contraction analysis in Theorem 2.2 (details in Appendix B.2). We write $\mathsf{CA}_{A \leftarrow B}(\mathbf{a}, \mathbf{b})$ for cross-attention with query $\mathbf{a}$ and key–value source $\mathbf{b}$:

$$\mathbf{o}_t^{(k+1)} = (1 - \alpha_o)\mathbf{o}_t^{(k)} + \alpha_o\, \mathsf{CA}_{\mathcal{O} \leftarrow \mathcal{M}}\big(\mathbf{o}_t^{(k)}, \mathbf{m}_t^{(k)}\big), \quad (1)$$

$$\mathbf{l}_t^{(k+1)} = (1 - \alpha_l)\mathbf{l}_t^{(k)} + \alpha_l\, \mathsf{CA}_{\mathcal{L} \leftarrow \mathcal{O}}\big(\mathbf{l}_t^{(k)}, \mathbf{o}_t^{(k+1)}\big), \quad (2)$$

$$\mathbf{m}_t^{(k+1)} = (1 - \alpha_m)\mathbf{m}_t^{(k)} + \alpha_m\, \mathsf{CA}_{\mathcal{M} \leftarrow \mathcal{L}}\big(\mathbf{m}_t^{(k)}, \mathbf{l}_t^{(k+1)}\big), \quad (3)$$

where $\alpha_o, \alpha_l, \alpha_m \in (0, 1)$ are relaxation weights. We use the Gauss–Seidel ordering (each update immediately feeds the next), and apply standard stability constraints inside $\mathsf{CA}$. (e.g., bounded-logit operation and spectral control), matching the assumptions of Theorem 2.2.

Let $\mathbf{z}^{(k)} = [\mathbf{o}^{(k)}; \mathbf{l}^{(k)}; \mathbf{m}^{(k)}]$ and denote the induced refinement operator by $T$. After $K$ refinement steps, we form the state-level control signal

$$\mathbf{C}_t = [\mathbf{o}_t^{(K)}; \mathbf{l}_t^{(K)}; \mathbf{m}_t^{(K)}], \quad (4)$$

which is injected into the frozen backbone to condition decoding (injection interface in §2.3).

**Refinement initialization.** We initialize $\mathbf{o}_t^{(0)}$ from pooled backbone states via $P_o(\mathrm{Pool}(\mathbf{H}_t))$, carry over memory as

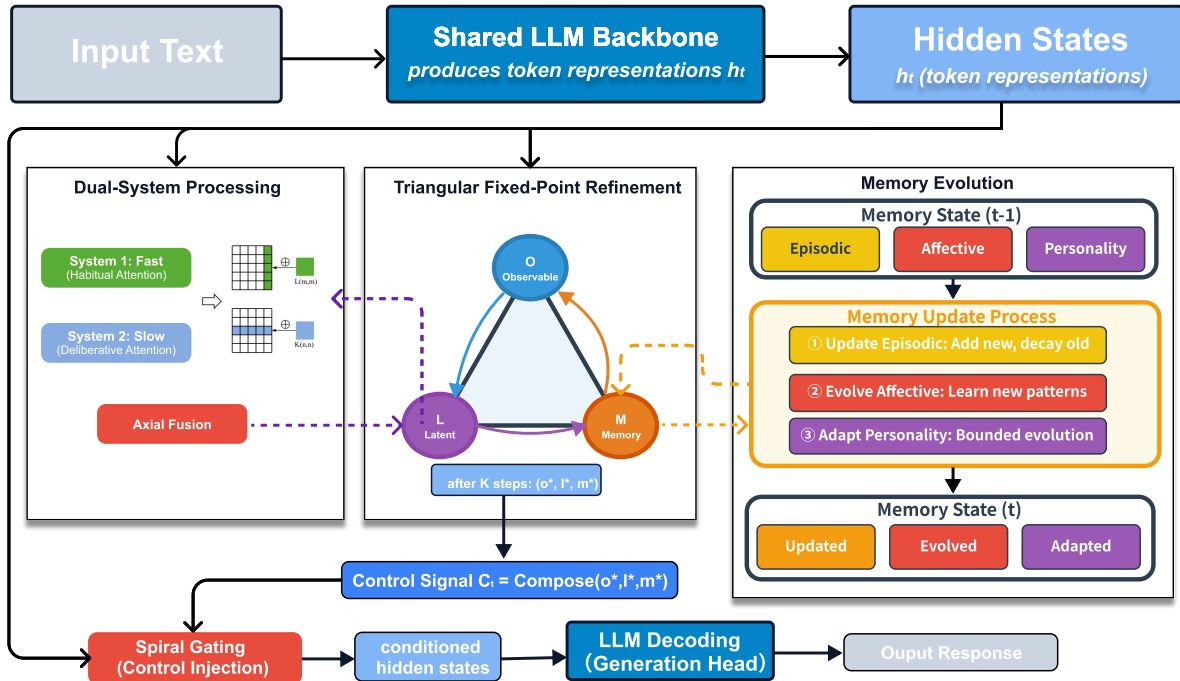

*Figure 1.* Overview of BRIDGE. The key insight is that Observable ($\mathcal{O}$), Latent ($\mathcal{L}$), and Memory ($\mathcal{M}$) must be *jointly* refined before decoding to prevent latent state drift. Triangular fixed-point refinement (center) achieves this with a provable convergence guarantee; tiered memory evolution (right) bounds long-horizon identity drift.

$\mathbf{m}_t^{(0)} = \tilde{\mathbf{m}}_{t-1}$, and set $\mathbf{l}_t^{(0)}$ using the dual-system fusion in Eq. 8. This yields a context-grounded start while preserving temporal continuity.

**Cyclical Coherence Loss.** When the sufficient contraction condition is only approximately satisfied, we add a soft alignment objective that penalizes post-refinement residual mismatch:

$$\mathcal{L}_{\text{cycle}} = \left\| \mathbf{o}_t^{(K)} - f_{\mathcal{O}}(\mathbf{m}_t^{(K)}) \right\|^2 + \left\| \mathbf{l}_t^{(K)} - f_{\mathcal{L}}(\mathbf{o}_t^{(K)}) \right\|^2$$
$$+ \left\| \mathbf{m}_t^{(K)} - f_{\mathcal{M}}(\mathbf{l}_t^{(K)}) \right\|^2,$$

(5)

where $f_{\mathcal{O}}, f_{\mathcal{L}}, f_{\mathcal{M}}$ are learned linear projections. This loss complements the sufficient condition in Eq. 6 by explicitly discouraging cross-space disagreement after refinement; further diagnostics are reported in Appendix B.2.

The following assumption and theorem establish convergence of the refinement operator.

**Assumption 2.1** (Bounded invariant operating domain). There exists a bounded set $\Omega \subset \mathbb{R}^{d_o + d_l + d_m}$ such that: (i) the refinement mapping $T$ induced by Eqs. (1)–(3) is well-defined on $\Omega$ and invariant (i.e., $T(\Omega) \subseteq \Omega$); and (ii) the pre-softmax logits inside each cross-attention block $\mathsf{CA}_{A \leftarrow B}$ remain uniformly bounded for all inputs in $\Omega$. Practical enforcement and audits of (i)–(ii) are provided in Appendix B.2.

**Theorem 2.2** (Domain-Restricted Convergence of Triangular Fixed-Point Refinement). *Under Assumption 2.1 and spectral control on the linear maps inside each* $\mathsf{CA}$., *let* $A \in \mathbb{R}_{\geq 0}^{3 \times 3}$ *denote the induced (nonnegative) coupling matrix for one Gauss–Seidel refinement step (explicit form in Appendix A.1). If*

$$\|A\|_{\infty} = \max_{i \in \{1,2,3\}} \sum_{j=1}^{3} A_{ij} < 1, \quad (6)$$

*then* $T : \Omega \to \Omega$ *is a contraction under the block max-norm* $\|\mathbf{z}\|_{\text{blk}\infty} \triangleq \max\{\|\mathbf{o}\|_2, \|\mathbf{l}\|_2, \|\mathbf{m}\|_2\}$. *Consequently, there exists a unique fixed point* $\mathbf{z}^\star \in \Omega$ *and the iterates satisfy* $\|\mathbf{z}^{(k)} - \mathbf{z}^\star\|_{\text{blk}\infty} \leq \|A\|_{\infty}^k \|\mathbf{z}^{(0)} - \mathbf{z}^\star\|_{\text{blk}\infty}$.

---

**Fixed-Point Convergence**

**Key condition.** $\|A\|_{\infty} < 1$ where $A$ is the Gauss–Seidel coupling matrix.

**Implication.** Unique fixed point; geometric convergence at rate $\|A\|_{\infty}^k$.

**Practical note.** Sufficient but not necessary; $\mathcal{L}_{\text{cycle}}$ provides a soft backup.

---

In practice, we set $\alpha_o = \alpha_l = \alpha_m = 0.25$ and $K = 3$.

## 2.3. Dual-System Processing

Triangular fixed-point refinement (§2.2) enforces *cross-space consistency*; the dual-system module complements it by providing the initial latent $\mathbf{l}_t^{(0)}$ and a turn-level steering signal $\mathbf{c}_t$, without modifying the refinement operator $T$.

**Seed latent.** To avoid circular dependencies, we construct a *seed latent* from backbone states, independent of refinement output:

$$\tilde{\mathbf{l}}_t = P_l\big(\mathrm{Pool}(\mathbf{H}_t)\big) \in \mathbb{R}^{d_l}, \tag{7}$$

where $\mathrm{Pool}(\cdot)$ is mean pooling over the sequence dimension and $P_l \in \mathbb{R}^{d_l \times d_h}$ is a learnable projection.

**Dual-system proposals.** We instantiate two complementary pathways along orthogonal axes: (i) a *horizontal* fast pathway $\mathbf{h}^{(1)}$ that queries a learned habitual key–value bank, and (ii) a *vertical* slow pathway $\mathbf{h}^{(2)}$ that applies a $D$-layer deliberative stack. Both are conditioned on the seed latent $\tilde{\mathbf{l}}_t$; parameterization details are in Appendix B.3.

**Axial Fusion.** Since the two pathways operate along different axes (horizontal retrieval vs. vertical depth), we fuse their outputs to produce the initial latent for triangular refinement:

$$\mathbf{l}_t^{(0)} = W_{\mathrm{fuse}}\big[\mathbf{h}^{(1)}; \mathbf{h}^{(2)}\big] + b_{\mathrm{fuse}}. \tag{8}$$

**Multiplicative Context-Aware Gating.** We inject control into the backbone by adaptively blending fast and slow pathway outputs. The gating network takes a multiplicative interaction term $\tilde{\mathbf{l}}_t \odot \mathbf{l}_t^{(0)}$ as input, enabling context–latent feature interleaving (a concise mapping from each named mechanism to its underlying operation is given in Appendix 3):

$$\mathbf{c}_t = g_t \odot \mathbf{h}^{(1)} + (1 - g_t) \odot \mathbf{h}^{(2)}, \tag{9}$$

where $g_t = \sigma\big(W_g[\tilde{\mathbf{l}}_t; \mathbf{l}_t^{(0)}; \tilde{\mathbf{l}}_t \odot \mathbf{l}_t^{(0)}] + b_g\big) \in [0,1]^{d_l}$. We inject both $\mathbf{c}_t$ (turn-level) and $\mathbf{C}_t$ (state-level, from §2.2) into the backbone:

$$\hat{\mathbf{H}}_t = \mathbf{H}_t + \Phi\big([\mathbf{c}_t; \mathbf{C}_t]\big), \tag{10}$$

where $\Phi(\cdot)$ projects to $\mathbb{R}^{d_h}$ and broadcasts along the sequence dimension.

## 2.4. Hierarchical Memory Evolution

Long-horizon persona consistency critically depends on *timescale separation*: recent episodes should update quickly, affective patterns should accumulate moderately, and core identity should evolve slowly. We therefore model memory as three tiers with distinct dynamics.

**Three-tier structure.** We maintain episodic, affective, and personality memory tiers with clear timescale separation: *episodic* ($\mathbf{m}^{(e)}$) stores recent interaction facts and events, *affective* ($\mathbf{m}^{(a)}$) accumulates emotional and relational patterns, and *personality* ($\mathbf{m}^{(p)}$) anchors slow-moving identity traits. Tier semantics and indicative timescales are reported in Appendix B.4.

**Interface with triangular refinement.** Triangular refinement operates on $\tilde{\mathbf{m}}_t$ within each turn. After decoding, each tier is updated via Eq. 11 with $\Delta\mathbf{m}_t^{(i)} \triangleq \mathbf{m}_t^{(i),(K)} - \mathbf{m}_{t-1}^{(i)}$; deployment-time snapshotting details are in Appendix B.4.

**Bounded Update Rules.** Each tier evolves via an anchored clipped update that explicitly references the initial persona encoding $\mathbf{m}_0^{(i)}$:

$$\mathbf{m}_{t+1}^{(i)} = (1-\eta_i)\mathbf{m}_t^{(i)} + \eta_i\left(\mathbf{m}_0^{(i)} + \mathrm{clip}\left(\Delta\mathbf{m}_t^{(i)}, -\delta_i, \delta_i\right)\right), \tag{11}$$

where $\eta_i \in \{\eta_e, \eta_a, \eta_p\}$ controls update speed and $\delta_i$ bounds the maximum per-step deviation. The design enforces two key properties: (i) *timescale separation* via $\eta_e > \eta_a > \eta_p$, ensuring that episodic details adapt quickly while personality evolves slowly; and (ii) *anchor regularization* via the $\mathbf{m}_0^{(i)}$ term, which pulls memory toward the initial persona when deviations are small, preventing unbounded drift. Together, these mechanisms enable the Lyapunov stability guarantee in Theorem 2.3.

**Lyapunov Stability Guarantee.** A key deployment concern is preventing *catastrophic drift*, where incremental deviations accumulate into identity changes. We provide a stability guarantee for bounded persona evolution.

**Theorem 2.3** (Lyapunov-Style Uniform Drift Bound for Persona Evolution). *Define* $V(\mathbf{m}_t) = \sum_{i\in\{e,a,p\}} \gamma_i \|\mathbf{m}_t^{(i)} - \mathbf{m}_0^{(i)}\|^2$ *with* $\gamma_i > 0$, *where* $\mathbf{m}_0^{(i)}$ *is the tier-$i$ anchor encoding of the initial persona,* $\mathbf{m}_{\mathrm{init}}^{(i)} \triangleq \mathbf{m}_{t=0}^{(i)}$ *denotes the tier-$i$ memory state at the start of the dialogue, and* $d_i = \dim(\mathbf{m}^{(i)})$. *Under the anchored clipped update in Eq. 11 with* $\eta_i \in (0,1)$ *and* $\delta_i > 0$, *for all* $t$,

$$V(\mathbf{m}_t) \leq \sum_{i\in\{e,a,p\}} \gamma_i\left(\max\{\|\mathbf{m}_{\mathrm{init}}^{(i)} - \mathbf{m}_0^{(i)}\|, \delta_i\sqrt{d_i}\}\right)^2. \tag{12}$$

*In particular, if the tier states are initialized at the persona anchor (i.e.,* $\mathbf{m}_{\mathrm{init}}^{(i)} = \mathbf{m}_0^{(i)}$), *then* $V(\mathbf{m}_t) \leq \sum_i \gamma_i \delta_i^2 d_i$ *for all* $t$.

> **Uniform Drift Bound**
>
> **Mechanism.** Anchored clipped updates pull memory toward initial persona $\mathbf{m}_0^{(i)}$.
>
> **Guarantee.** $V(\mathbf{m}_t) \le V_{\max}$ for all $t$; drift controlled by $\delta_i$.
>
> **Design knob.** Small $(\eta_p, \delta_p)$ anchors identity; larger $(\eta_e, \delta_e)$ enables fast adaptation.

This result formalizes *adaptation with bounded drift*: the timescale hierarchy $\eta_e > \eta_a > \eta_p$ ensures that episodic details adapt quickly while personality evolves slowly.

## 2.5. Training Objective

We optimize BRIDGE using parameter-efficient fine-tuning: the pretrained LLM backbone (Qwen2.5-32B-Instruct) remains *frozen*, and only the BRIDGE-specific modules are trained. These include the triangular cross-attention layers, dual-system pathways, memory projections, the multiplicative context-aware gating, and the injection layer $\Phi$ (detailed in §2.2–§2.4). This results in **277M trainable parameters** (Appendix B.1), keeping BRIDGE parameter-efficient relative to the frozen backbone.

The composite training objective is:

$$\mathcal{L}_{\text{total}} = \mathcal{L}_{\text{LM}} + \lambda_1 \mathcal{L}_{\text{cycle}} + \lambda_2 \mathcal{L}_{\text{persona}}, \qquad (13)$$

where $\mathcal{L}_{\text{LM}}$ is the next-token loss computed through the frozen LM head, $\mathcal{L}_{\text{cycle}}$ is the triangular coherence loss (Eq. 5), and $\mathcal{L}_{\text{persona}}$ is a BCE loss from a lightweight persona-consistency classifier (Appendix B.5). Gradients from all three terms flow through the frozen backbone to update BRIDGE parameters. We set $\lambda_1 = 0.1$ and $\lambda_2 = 0.5$ on validation.

**Algorithm summary.** We summarize the complete BRIDGE inference procedure in Algorithm 1. The algorithm executes four stages per dialogue turn: (I) state initialization via dual-system processing, (II) triangular fixed-point refinement for cross-space consistency, (III) control injection and response decoding, and (IV) hierarchical memory evolution. For deployment, working memory is periodically snapshotted to persistent storage (Appendix B.4).

## 3. Experiments

We evaluate BRIDGE on persona consistency benchmarks to answer three questions: (1) Does explicit $\mathcal{O}$–$\mathcal{L}$–$\mathcal{M}$ coupling improve persona fidelity over strong baselines? (2) Which components contribute most, and what mechanisms underlie their effectiveness? (3) Do the theoretical guarantees (Theorems 2.2–2.3) hold empirically? We describe setup in §3.1, present main results in §3.2, provide ablations

---

**Algorithm 1** BRIDGE inference workflow (per dialogue turn $t$).

---

**Require:** Dialogue context $(\mathcal{H}, u_t)$, initial persona $P_0$, frozen backbone $\theta_{\text{LM}}$, previous memory $\tilde{\mathbf{m}}_{t-1}$
**Ensure:** Response $r_t$, updated memory $\mathbf{m}_t$
1: $\mathbf{H}_t \leftarrow \text{Backbone}_\theta(\text{context})$ ▷ frozen encoding
2: */\* Stage I: State Initialization \*/*
3: $\tilde{\mathbf{l}}_t \leftarrow P_l(\text{Pool}(\mathbf{H}_t))$ ▷ Eq. 7
4: $\mathbf{h}_t^{(1)} \leftarrow \text{Attn}_{\text{fast}}(\tilde{\mathbf{l}}_t, \mathbf{K}_{\text{habit}}, \mathbf{V}_{\text{habit}})$ ▷ System 1
5: $\mathbf{h}_t^{(2)} \leftarrow \text{Attn}_{\text{slow}}^{(D)} \circ \cdots \circ \text{Attn}_{\text{slow}}^{(1)}(\tilde{\mathbf{l}}_t)$ ▷ System 2
6: $\mathbf{l}_t^{(0)} \leftarrow W_{\text{fuse}}[\mathbf{h}_t^{(1)}; \mathbf{h}_t^{(2)}] + b_{\text{fuse}}$ ▷ Eq. 8
7: $\mathbf{o}_t^{(0)} \leftarrow P_o(\text{Pool}(\mathbf{H}_t)); \quad \mathbf{m}_t^{(0)} \leftarrow \tilde{\mathbf{m}}_{t-1}$
8: */\* Stage II: Triangular Fixed-Point Refinement \*/*
9: **for** $k = 0$ to $K - 1$ **do**
10: $\quad \mathbf{o}_t^{(k+1)} \leftarrow (1 - \alpha_o)\mathbf{o}_t^{(k)} + \alpha_o \cdot \text{CA}_{\mathcal{O} \leftarrow \mathcal{M}}(\mathbf{o}_t^{(k)}, \mathbf{m}_t^{(k)})$
$\quad$ ▷ Eq. 1
11: $\quad \mathbf{l}_t^{(k+1)} \leftarrow (1 - \alpha_l)\mathbf{l}_t^{(k)} + \alpha_l \cdot \text{CA}_{\mathcal{L} \leftarrow \mathcal{O}}(\mathbf{l}_t^{(k)}, \mathbf{o}_t^{(k+1)})$
$\quad$ ▷ Eq. 2
12: $\quad \mathbf{m}_t^{(k+1)} \leftarrow (1 - \alpha_m)\mathbf{m}_t^{(k)} + \alpha_m \cdot \text{CA}_{\mathcal{M} \leftarrow \mathcal{L}}(\mathbf{m}_t^{(k)}, \mathbf{l}_t^{(k+1)})$ ▷ Eq. 3
13: **end for**
14: */\* Stage III: Control Injection & Decoding \*/*
15: $\mathbf{C}_t \leftarrow [\mathbf{o}_t^{(K)}; \mathbf{l}_t^{(K)}; \mathbf{m}_t^{(K)}]$ ▷ Eq. 4
16: $g_t \leftarrow \sigma(W_g[\tilde{\mathbf{l}}_t; \mathbf{l}_t^{(0)}; \tilde{\mathbf{l}}_t \odot \mathbf{l}_t^{(0)}] + b_g)$
17: $\mathbf{c}_t \leftarrow g_t \odot \mathbf{h}_t^{(1)} + (1 - g_t) \odot \mathbf{h}_t^{(2)}$ ▷ Eq. 9
18: $\hat{\mathbf{H}}_t \leftarrow \mathbf{H}_t + \Phi([\mathbf{c}_t; \mathbf{C}_t])$ ▷ Eq. 10
19: $r_t \leftarrow \text{Decode}(\hat{\mathbf{H}}_t)$
20: */\* Stage IV: Hierarchical Memory Evolution \*/*
21: **for** $i \in \{e, a, p\}$ **do**
22: $\quad \Delta\mathbf{m}_t^{(i)} \leftarrow \mathbf{m}_t^{(i),(K)} - \mathbf{m}_{t-1}^{(i)}$
23: $\quad \mathbf{m}_{t+1}^{(i)} \leftarrow (1 - \eta_i)\mathbf{m}_t^{(i)} + \eta_i(\mathbf{m}_0^{(i)} + \text{clip}(\Delta\mathbf{m}_t^{(i)}, -\delta_i, \delta_i))$ ▷ Eq. 11
24: **end for**

---

and mechanistic analysis in §3.3, and quantify asymmetric rupture risk in §3.4.

### 3.1. Experimental Setup

**Benchmarks.** We evaluate on two complementary benchmarks. **PersonaGym** (Samuel et al., 2024) provides single-turn interview-style evaluation over 200 personas and 10K prompts, reporting five dimensions: Action Justification (AJ), Expected Action (EA), Linguistic Habits (LH), Persona Consistency (PC), and Toxicity Control (TC), with *PersonaScore* as their unweighted average. **CoSER** (Wang et al., 2025a) evaluates multi-turn given-circumstance acting over 200 conversations (3 characters; 18 turns), reporting Storyline Consistency (SC), Anthropomorphism (An), Character Fidelity (CF), and Storyline Quality (SQ). We further

*Table 1.* Performance on PersonaGym (single-turn) and CoSER (multi-turn). BRIDGE achieves the best average scores on both benchmarks. Results are mean ± std over 3 runs; GPT-4o as judge. **Bold** indicates best per column.

| Method | PersonaGym | | | | | | CoSER | | | | |
| --- | --- | --- | --- | --- | --- | --- | --- | --- | --- | --- | --- |
| | AJ | EA | LH | PC | TC | Avg. | SC | An | CF | SQ | Avg. |
| *Zero-Shot Prompting* | | | | | | | | | | | |
| Qwen3-8B | 3.95±0.04 | 3.80±0.05 | 4.02±0.03 | 4.10±0.04 | 4.72±0.02 | 4.12±0.03 | 56.3±1.2 | 44.7±1.5 | 39.5±1.8 | 64.8±1.1 | 51.3±1.0 |
| Qwen2.5-32B-Instruct | 4.28±0.03 | 4.08±0.04 | 3.95±0.03 | 4.45±0.03 | 4.80±0.02 | 4.31±0.02 | 55.5±1.0 | 45.2±1.3 | 40.2±1.5 | 72.8±0.9 | 53.4±0.8 |
| LLaMA-3.3-70B | 4.22±0.03 | 4.05±0.04 | 3.90±0.04 | 4.40±0.03 | 4.78±0.02 | 4.27±0.02 | 55.0±1.1 | 46.1±1.4 | 36.4±1.6 | 73.4±1.0 | 52.7±0.9 |
| DeepSeek-v3.2 | 4.33±0.03 | 4.20±0.03 | 4.18±0.03 | 4.55±0.03 | 4.55±0.03 | 4.36±0.02 | 57.2±1.0 | 45.6±1.2 | 40.8±1.4 | 71.5±1.1 | 53.8±0.8 |
| GPT-4o | 4.50±0.02 | 4.18±0.03 | 4.06±0.03 | 4.64±0.02 | 4.93±0.01 | 4.46±0.02 | 59.4±0.8 | 44.2±1.1 | 42.1±1.3 | 76.0±0.8 | 55.4±0.7 |
| Claude-3.7-Sonnet | 4.47±0.02 | 4.29±0.03 | 4.10±0.03 | 4.62±0.02 | **4.95**±0.01 | 4.49±0.02 | 60.1±0.9 | 47.8±1.0 | 43.0±1.2 | 74.8±0.9 | 56.4±0.7 |
| *Chain-of-Thought Prompting* | | | | | | | | | | | |
| Qwen2.5-32B-Instruct + CoT | 4.35±0.03 | 4.15±0.03 | 4.02±0.03 | 4.52±0.03 | 4.82±0.02 | 4.37±0.02 | 56.8±1.0 | 46.5±1.2 | 41.8±1.4 | 73.5±0.9 | 54.7±0.8 |
| GPT-4o + CoT | 4.55±0.02 | 4.25±0.03 | 4.12±0.03 | 4.68±0.02 | 4.94±0.01 | 4.51±0.02 | 60.5±0.8 | 45.8±1.0 | 43.5±1.2 | 76.8±0.8 | 56.7±0.6 |
| *Specialized Persona Methods* | | | | | | | | | | | |
| Character-LLM | 3.98±0.05 | 3.82±0.05 | 3.58±0.06 | 4.18±0.04 | 4.55±0.03 | 4.02±0.04 | 53.5±1.4 | 44.8±1.6 | 40.2±1.7 | 62.8±1.2 | 50.3±1.2 |
| *Parameter-Efficient Fine-Tuning (Qwen2.5-32B-Instruct backbone)* | | | | | | | | | | | |
| LoRA (r=64) | 4.42±0.03 | 4.22±0.03 | 4.08±0.03 | 4.58±0.03 | 4.81±0.02 | 4.42±0.02 | 58.2±1.1 | 47.5±1.3 | 43.8±1.4 | 73.2±1.0 | 55.7±0.9 |
| Adapter | 4.40±0.03 | 4.20±0.03 | 4.05±0.04 | 4.55±0.03 | 4.79±0.02 | 4.40±0.02 | 57.8±1.2 | 47.2±1.4 | 43.5±1.5 | 73.0±1.1 | 55.4±0.9 |
| Neeko (Dynamic LoRA) | 4.45±0.03 | 4.25±0.03 | 4.12±0.03 | 4.60±0.03 | 4.78±0.02 | 4.44±0.02 | 58.5±1.0 | 48.0±1.2 | 45.5±1.3 | 73.8±1.0 | 56.5±0.8 |
| BRIDGE (Ours) | **4.65**±0.02 | **4.42**±0.02 | **4.28**±0.02 | **4.82**±0.02 | 4.76±0.02 | **4.59**±0.02 | **63.8**±0.9 | **52.4**±1.0 | **48.2**±1.1 | 73.5±0.9 | **59.5**±0.7 |

use **LoCoMo** (Maharana et al., 2024) for long-horizon stability diagnostics (Appendix D.1). Full protocols are in Appendix C.

**Baselines.** We compare against (i) open-source LLMs (Qwen3-8B, Qwen2.5-32B-Instruct (Yang et al., 2025), LLaMA-3.3-70B (Dubey et al., 2024), DeepSeek-v3.2 (Liu et al., 2025a)) and proprietary models (GPT-4o (Hurst et al., 2024), Claude-3.7-Sonnet) under zero-shot persona-in-context prompting, (ii) Character-LLM (Shao et al., 2023) as a representative specialized persona method, (iii) backbone-matched PEFT (Xu et al., 2023) baselines (LoRA, Adapter, and Neeko's dynamic LoRA (Yu et al., 2024)) on Qwen2.5-32B-Instruct, and (iv) CoT prompting to isolate inference-time deliberation from architectural coupling.

### 3.2. Main Results

Table 1 reports results on PersonaGym and CoSER. We highlight three takeaways: (i) BRIDGE achieves the best overall performance, including under backbone-matched comparisons; (ii) improvements concentrate on persona-fidelity dimensions rather than generic storytelling quality; and (iii) Toxicity Control changes only marginally.

**BRIDGE achieves the best overall performance.** BRIDGE attains the highest averages on both benchmarks (PersonaGym: 4.59±0.02; CoSER: 59.5±0.7). Under backbone-matched comparison on Qwen2.5-32B-Instruct, BRIDGE surpasses strong PEFT baselines (LoRA/Adapter), indicating that the gains stem from explicit $\mathcal{O}$–$\mathcal{L}$–$\mathcal{M}$ coupling rather than additional trainable parameters alone. This coupling-driven gain is not backbone-specific: on a different architecture family (LLaMA-3-8B-Instruct), BRIDGE again outperforms a parameter-matched LoRA baseline with the

same persona-concentrated fingerprint (Appendix D.5).

CoT prompting improves performance but remains consistently below BRIDGE, suggesting that inference-time deliberation is complementary yet insufficient to replace persistent state reconciliation.

**Gains concentrate on persona fidelity rather than generic storytelling.** On PersonaGym, the largest gains are Persona Consistency and Action Justification; on CoSER, gains peak on Character Fidelity and Anthropomorphism, while Storyline Quality changes only modestly. This selective pattern supports our mechanism claim: BRIDGE reduces long-horizon persona drift by reconciling behavior, latent cognition, and memory *before* decoding, rather than broadly increasing narrative fluency. Toxicity Control slightly decreases (−0.19 vs. Claude-3.7-Sonnet), reflecting rubric tension: faithful portrayal of confrontational personas can trigger toxicity detectors even when contextually appropriate (Appendix C).

### 3.3. Ablations and Mechanistic Validation

To understand how BRIDGE mitigates persona drift, we use *ablations* and *mechanistic probes*. Across PersonaGym and CoSER, the largest drops come from weakening either the *identity anchor* (hierarchical memory evolution, especially the personality tier) or the *closed-loop reconciliation* (Triangular Fixed-Point Refinement), while secondary modules (dual-system routing and additional refinement steps) yield smaller but consistent gains.

**Hierarchical memory provides a stable long-horizon identity anchor.** Ablating hierarchical memory evolution causes the most substantial performance decline, and tier ablations identify *personality memory* as the primary anchor

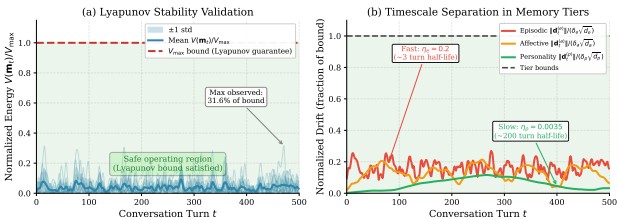

*Figure 2.* Theorem 2.3 validated on **LoCoMo** (500-turn auto-regressive simulation). **(a)** $V(\mathbf{m}_t)/V_{\max}$ stays below the theoretical bound (dashed), with a peak at 31.6% of the bound. **(b)** Timescale separation in tier-wise drift: episodic ($\eta_e = 0.2$, $\sim$3–4 turns) vs. personality ($\eta_p = 0.0035$, $\sim$200 turns), both within bounds.

for long-horizon identity (w/o personality memory: $-6.1$ on CoSER). To mechanistically validate that this anchor prevents unbounded drift, we track the Lyapunov energy $V(\mathbf{m}_t)$ induced by our anchored clipped update (Eq. 11) on long-context conversations from **LoCoMo** over 500 turns. Figure 2(a) shows that normalized energy $V(\mathbf{m}_t)/V_{\max}$ remains well below the theoretical bound, peaking at 31.6% of it, consistent with Theorem 2.3 as a conservative stability guarantee. Figure 2(b) further confirms timescale separation: episodic memory adapts rapidly to recent events, while personality memory evolves slowly, consistent with its role as a persistent identity reference. Interpretable Big-5 personality trait proxies derived from $\mathbf{m}_t^{(p)}$ exhibit similar bounded trajectories (Appendix D.1, Figure 5).

**Closed-loop triangular reconciliation is necessary, and it aligns functionally specialized states.** Replacing Triangular Fixed-Point Refinement with standard attention yields the largest performance drop, while partial couplings (only $\mathcal{O} \leftrightarrow \mathcal{L}$ or only $\mathcal{L} \leftrightarrow \mathcal{M}$) underperform consistently. Using a single refinement step ($K=1$) yields a smaller yet consistent drop, consistent with Theorem 2.2: iterative refinement reduces cross-space disagreement via geometric contraction of residual mismatch toward a fixed point, beyond a one-pass update.

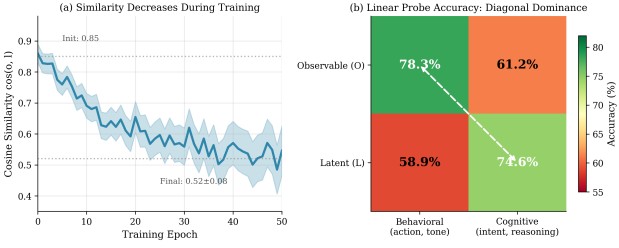

*Figure 3.* Observable–Latent semantic differentiation. **(a)** Cosine similarity $\cos(\mathbf{o}, \mathbf{l})$ drops from 0.85 at init to $0.52 \pm 0.08$ after training (mean ± std, 3 runs). **(b)** Linear probe accuracy (%): diagonal dominance (O→behavioral, L→cognitive) indicates functional specialization of O and L.

We address the concern that Observable (O) and Latent (L) could be redundant. Figure 3 argues against representational collapse: cosine similarity drops substantially after training, and linear probes show matched-task superiority— O predicts *behavioral* attributes more accurately (78.3% vs. 58.9%), L predicts *cognitive* attributes more accurately (74.6% vs. 61.2%)—indicating functional specialization rather than collapse.

*Table 2.* Ablation study on PersonaGym and CoSER. Each configuration removes or modifies one architectural component. $\Delta$ shows absolute change from Full BRIDGE.

| Configuration | PersonaGym | | CoSER | |
|---|---|---|---|---|
| | Score | $\Delta$ | Score | $\Delta$ |
| Full BRIDGE | 4.59 | — | 59.5 | — |
| ***Core Components*** | | | | |
| w/o Triangular Fixed-Point Refinement | 4.49 | $-0.10$ | 56.8 | $-2.7$ |
| w/o Dual-System Processing | 4.53 | $-0.06$ | 57.9 | $-1.6$ |
| w/o Memory Evolution | 4.45 | $-0.14$ | 55.2 | $-4.3$ |
| ***Coupling Mechanism Variants*** | | | | |
| Standard Attention | 4.40 | $-0.19$ | 54.6 | $-4.9$ |
| Only $\mathcal{O} \leftrightarrow \mathcal{L}$ | 4.50 | $-0.09$ | 56.5 | $-3.0$ |
| Only $\mathcal{L} \leftrightarrow \mathcal{M}$ | 4.48 | $-0.11$ | 57.1 | $-2.4$ |
| No Fixed-Point ($K = 1$) | 4.55 | $-0.04$ | 58.3 | $-1.2$ |
| ***Dual-System Variants*** | | | | |
| Only System 1 (Fast) | 4.45 | $-0.14$ | 56.2 | $-3.3$ |
| Only System 2 (Slow) | 4.49 | $-0.10$ | 57.4 | $-2.1$ |
| Linear Combination | 4.48 | $-0.11$ | 56.8 | $-2.7$ |
| ***Memory Tier Variants*** | | | | |
| w/o Episodic Memory | 4.53 | $-0.06$ | 57.6 | $-1.9$ |
| w/o Affective Memory | 4.51 | $-0.08$ | 56.8 | $-2.7$ |
| w/o Personality Memory | 4.35 | $-0.24$ | 53.4 | $-6.1$ |
| w/o Memory Decay | 4.48 | $-0.11$ | 55.9 | $-3.6$ |
| Only Episodic Memory | 4.30 | $-0.29$ | 52.1 | $-7.4$ |
| Only Affective Memory | 4.21 | $-0.38$ | 50.8 | $-8.7$ |
| Only Personality Memory | 4.34 | $-0.25$ | 53.8 | $-5.7$ |

**Dual-system routing enables adaptive computation.** Dual-system processing provides complementary gains: using only the fast or only the slow pathway underperforms the learned router, and a naive linear combination is inferior (Table 2). This pattern suggests the router adaptively selects habitual versus deliberative computation based on context, rather than merely adding capacity.

Additional analyses (longer-run diagnostics, memory activation dynamics, cross-space coherence, and cross-judge validation) are in Appendix D.

### 3.4. Asymmetric Rupture Risk

Beyond aggregate metrics, persona dialogue systems face *asymmetric rupture risk*: the cost of a single severe contradiction far exceeds the benefit of many locally coherent exchanges. This asymmetry is well-documented—users tolerate gradual degradation but abandon systems after a single salient error (Dietvorst et al., 2015), with negative events carrying approximately twice the psychological weight of positive ones (Kahneman & Tversky, 2013). In persona dialogue, one severe contradiction—an amnesiac detective sud-

denly recalling vivid childhood memories—can irreversibly rupture user trust regardless of preceding coherent turns.

We operationalize rupture as conversations where Character Fidelity falls below 35%, capturing severe contradiction rather than minor drift (threshold sensitivity in Appendix D.7). The key finding is a striking *amplification effect*: on a stratified subset of 100 CoSER dialogues, BRIDGE reduces rupture rate from 37% to 15%—a **59% relative reduction** from only +8.0 mean CF improvement. This disproportionate tail-risk reduction emerges directly from the triangular coupling: the geometric contraction in Theorem 2.2 bounds worst-case deviations that baselines allow to compound into catastrophic contradiction.

This result connects to the ablation findings in § 3.3: removing Triangular Fixed-Point Refinement causes the largest performance drop because it eliminates the contraction bounding worst-case trajectories. Partial couplings reduce mean error but leave subspaces where drift compounds undetected; full O–L–M closure converts potential ruptures into bounded deviations caught during reconciliation. Qualitative failure modes are illustrated in Appendix D.6; distribution analysis in Appendix D.7.

## 4. Related Work

**Persona-Grounded Dialogue.** Persona conditioning has evolved from static profile-based generation (Zhang et al., 2018) to dynamic paradigms, including implicit persona extraction (Liu et al., 2020), commonsense-augmented personas (Majumder et al., 2020), and multi-turn consistency mechanisms (Kim et al., 2020). Recent work strengthens role fidelity through experience-based tuning (Shao et al., 2023; Wang et al., 2024) and parameter-efficient adaptation: Neeko (Yu et al., 2024) uses dynamic LoRA blocks with MoE-style gating; PCL (Ji et al., 2025) combines persona-aware contrastive learning with self-play alignment; and ID-RAG (Platnick et al., 2025) employs identity knowledge graphs to reduce drift. Evaluation has also advanced, with PersonaGym (Samuel et al., 2024) providing scenario-based testing and CoSER (Wang et al., 2025a) introducing given-circumstance acting. Despite this progress, most systems treat persona as a static conditioning signal, decoupling behavior from belief and memory. BRIDGE addresses this by explicitly modeling the triangular O–L–M dependencies.

**Memory Architectures for Long-Horizon Dialogue.** Long-term dialogue requires memory mechanisms beyond fixed context windows. Retrieval-augmented generation (Lewis et al., 2020) enables access to external knowledge but treats memory as a flat store; MemGPT (Packer et al., 2023) introduces hierarchical memory management; and MemoryBank (Zhong et al., 2024) incorporates forgetting dynamics inspired by psychological retention curves.

More recent approaches employ co-evolving world models for state prediction (Fang et al., 2025b) or Markovian workspace reconstruction for unbounded exploration (Chen et al., 2025). Benchmarking efforts such as LoCoMo (Maharana et al., 2024) reveal persistent gaps in long-horizon coherence even for strong LLMs. These approaches scale memory capacity or retrieval sophistication but lack explicit drift constraints. BRIDGE differs by coupling memory updates to persona evolution across multiple timescales and enforcing anchored clipped updates with a Lyapunov-style drift bound (Theorem 2.3).

**Cognitively Inspired Processing and Structured Attention.** Dual-process accounts distinguish fast, automatic responses from slow, deliberative reasoning (Kahneman, 2011), formalized in classical cognitive architectures such as ACT-R (Anderson et al., 2004) and SOAR (Laird, 2019). Recent work applies this framing to LLMs (Hagendorff et al., 2023; Bengio et al., 2019), often via prompting strategies like chain-of-thought (Wei et al., 2022) rather than architectural primitives. In hierarchical RL, value function spaces (Shah et al., 2021) use skill-centric abstractions for long-horizon reasoning. Separately, higher-order attention mechanisms have been explored: the 2-simplicial Transformer (Clift et al., 2019) generalizes attention to trilinear forms; AlphaFold2 (Jumper et al., 2021) uses triangle attention to enforce geometric consistency; and the Edge Transformer (Bergen et al., 2021) updates edge representations via triangular aggregation. BRIDGE combines both lines: parallel fast/slow pathways with learned multiplicative context-aware gating, and a closed-loop pairwise coupling $\mathcal{M} \to \mathcal{O} \to \mathcal{L} \to \mathcal{M}$ with per-edge spectral control enabling the contraction guarantee (Theorem 2.2).

**Companion Work.** A companion ICML 2026 paper by the same authors, KSKT (Sun et al., 2026), addresses a complementary failure mode: *within-turn role–user conflict*, via attention factorization enabling token-level dual-perspective arbitration. BRIDGE targets *cross-turn* drift— the silent divergence among behavior, belief, and memory that compounds over sessions. The two operate at non-overlapping timescales and compose naturally: KSKT can serve as the per-turn generator within BRIDGE's session-level state loop.

## 5. Conclusion

We introduced BRIDGE, a framework that explicitly couples Observable behaviors ($\mathcal{O}$), Latent mental states ($\mathcal{L}$), and Memory ($\mathcal{M}$) through triangular fixed-point refinement with provable convergence guarantees. Combined with dual-process computation via multiplicative context-aware gating and hierarchical memory evolution with Lyapunov-bounded updates, BRIDGE achieves state-of-the-art performance

on PersonaGym and CoSER, with gains concentrated on persona-specific metrics while generic storytelling quality remains stable. This validates our core claim: explicit $\mathcal{O}$–$\mathcal{L}$–$\mathcal{M}$ coupling enables *adaptation with bounded drift*, addressing the structural bottleneck underlying long-horizon persona inconsistency. We hope this work demonstrates that principled architectural constraints, grounded in convergence and stability theory, can yield practical gains in deployed dialogue systems.

Limitations include reliance on LLM judges ($\rho \geq 0.91$ cross-judge), partially mitigated by a preliminary human evaluation that reproduces the automatic ranking (Appendix C.3); a full-scale human study remains future work. Coverage of specialized persona methods is also limited (only Character-LLM). Future work includes drift analysis under distribution shift, extensions to task-adaptive instruction-following, and further latency optimization of the coupling modules.

## Impact Statement

This work targets long-horizon persona consistency in dialogue agents, with potential benefits for trust-sensitive applications such as education and supportive conversations. By enforcing cross-space reconciliation and bounding memory-driven persona evolution, BRIDGE can reduce high-cost contradictions across sessions and improve interaction predictability. At the same time, more stable persona behavior may increase the risk of inappropriate user reliance or dependency in companion-like deployments; we therefore recommend clear disclosure, calibrated expectation setting, and human-in-the-loop oversight for any mental-health-related use. Moreover, because hierarchical memory can encode user interaction patterns, deployments should minimize retention by default, apply privacy-preserving storage and access controls, and avoid collecting or storing sensitive attributes unless strictly necessary. Finally, BRIDGE is not a substitute for licensed professional services; for high-stakes domains, it should be deployed only with domain-specific safety audits, monitoring, and escalation pathways.

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

# A. Theoretical Proofs

This appendix provides complete proofs for the theoretical results in the main text. For readability, we restate each theorem before its proof.

## A.1. Proof of Theorem 2.2 (Convergence of Triangular Fixed-Point Refinement)

**Theorem A.1** (Restated: Domain-Restricted Convergence of Triangular Fixed-Point Refinement). *Let $\sigma(\cdot)$ denote spectral normalization such that $\sigma(W) \leq 1$ for all weight matrices $W$ used inside each cross-attention block $\mathsf{CA}_{\cdot\cdot}$. Under Assumption 2.1, suppose the cross-attention blocks admit finite domain-restricted Lipschitz bounds as in Eq. (14). Let $A \in \mathbb{R}_{\geq 0}^{3 \times 3}$ be the induced nonnegative coupling matrix (Eq. (21)).*

*Define the block max-norm on joint states $\mathbf{z} = [\mathbf{o}; \mathbf{l}; \mathbf{m}]$ by*

$$\|\mathbf{z}\|_{\mathrm{blk}\infty} \triangleq \max\{\|\mathbf{o}\|_2, \|\mathbf{l}\|_2, \|\mathbf{m}\|_2\}.$$

*If $\|A\|_\infty < 1$, then the joint refinement operator $T : \Omega \to \Omega$ is a contraction on $\Omega$ under $\|\cdot\|_{\mathrm{blk}\infty}$, and hence admits a unique fixed point in $\Omega$ with geometric convergence from any initialization in $\Omega$.*

*Proof.* We prove convergence on the bounded invariant set $\Omega$ by establishing that the induced joint operator is a contraction under the block max-norm $\|\cdot\|_{\mathrm{blk}\infty}$.

**Joint state and operator.** Let the joint state be $\mathbf{z} \triangleq [\mathbf{o}; \mathbf{l}; \mathbf{m}] \in \mathbb{R}^{d_o + d_l + d_m}$. Define the one-step refinement mapping induced by Equations (1)–(3) as $T : \Omega \to \Omega$, i.e., $\mathbf{z}^{(k+1)} = T(\mathbf{z}^{(k)})$.

**Norm conventions.** Throughout this proof, $\|\cdot\|$ denotes the Euclidean norm on vector blocks (e.g., $\mathbf{o} \in \mathbb{R}^{d_o}$, $\mathbf{l} \in \mathbb{R}^{d_l}$, $\mathbf{m} \in \mathbb{R}^{d_m}$), and for $\Delta \in \mathbb{R}^3$ we use $\|\Delta\|_\infty$ to denote the vector $\ell_\infty$ norm, while $\|A\|_\infty$ denotes the induced matrix $\ell_\infty$ norm defined in Eq. (6).

**Notation for Lipschitz bounds.** For each cross-attention block, write $\mathsf{CA}_{A \leftarrow B}(\mathbf{a}, \mathbf{b})$ where $\mathbf{a}$ is the query-side input (space $A$) and $\mathbf{b}$ is the key/value-side input (space $B$). On $\Omega$, define finite constants $L_{A \leftarrow B}^{(a)}$ and $L_{A \leftarrow B}^{(b)}$ such that

$$\left\|\mathsf{CA}_{A \leftarrow B}(\mathbf{a}_1, \mathbf{b}_1) - \mathsf{CA}_{A \leftarrow B}(\mathbf{a}_2, \mathbf{b}_2)\right\| \leq L_{A \leftarrow B}^{(a)} \|\mathbf{a}_1 - \mathbf{a}_2\| + L_{A \leftarrow B}^{(b)} \|\mathbf{b}_1 - \mathbf{b}_2\| \tag{14}$$

for all $(\mathbf{a}_1, \mathbf{b}_1), (\mathbf{a}_2, \mathbf{b}_2) \in \Omega$. Such bounds are well-defined on bounded domains for dot-product attention; see Kim et al. (2021); Dasoulas et al. (2021). Moreover, sharper constants follow from tight Lipschitz bounds of softmax on bounded logits (Nair, 2025). Spectral normalization ensures the linear projections inside each attention block have operator norm at most 1, so under bounded logits (Assumption 2.1(ii)) the remaining softmax nonlinearity admits a finite domain-restricted Lipschitz constant, yielding finite $L_{A \leftarrow B}^{(\cdot)}$ on $\Omega$.

**Step 1: Component-wise contraction under relaxed updates.** Consider two joint states $\mathbf{z}_1 = [\mathbf{o}_1; \mathbf{l}_1; \mathbf{m}_1]$ and $\mathbf{z}_2 = [\mathbf{o}_2; \mathbf{l}_2; \mathbf{m}_2]$ in $\Omega$. From Equation (1) and (14),

$$\|\mathbf{o}_1^+ - \mathbf{o}_2^+\| = \left\|(1 - \alpha_o)(\mathbf{o}_1 - \mathbf{o}_2) + \alpha_o\big(\mathsf{CA}_{\mathcal{O} \leftarrow \mathcal{M}}(\mathbf{o}_1, \mathbf{m}_1) - \mathsf{CA}_{\mathcal{O} \leftarrow \mathcal{M}}(\mathbf{o}_2, \mathbf{m}_2)\big)\right\|$$
$$\leq \big((1 - \alpha_o) + \alpha_o L_{\mathcal{O} \leftarrow \mathcal{M}}^{(o)}\big)\|\mathbf{o}_1 - \mathbf{o}_2\| + \alpha_o L_{\mathcal{O} \leftarrow \mathcal{M}}^{(m)}\|\mathbf{m}_1 - \mathbf{m}_2\|. \tag{15}$$

Similarly, using Equation (2) with $\mathbf{o}^+$ as input,

$$\|\mathbf{l}_1^+ - \mathbf{l}_2^+\| \leq \big((1 - \alpha_l) + \alpha_l L_{\mathcal{L} \leftarrow \mathcal{O}}^{(l)}\big)\|\mathbf{l}_1 - \mathbf{l}_2\| + \alpha_l L_{\mathcal{L} \leftarrow \mathcal{O}}^{(o)}\|\mathbf{o}_1^+ - \mathbf{o}_2^+\|. \tag{16}$$

Finally, for Equation (3),

$$\|\mathbf{m}_1^+ - \mathbf{m}_2^+\| \leq \big((1 - \alpha_m) + \alpha_m L_{\mathcal{M} \leftarrow \mathcal{L}}^{(m)}\big)\|\mathbf{m}_1 - \mathbf{m}_2\| + \alpha_m L_{\mathcal{M} \leftarrow \mathcal{L}}^{(l)}\|\mathbf{l}_1^+ - \mathbf{l}_2^+\|. \tag{17}$$

**Step 2: A coupled inequality and a block-$\ell_\infty$ contraction.** For brevity, we write $\mathbf{o}^+ \triangleq \mathbf{o}^{(k+1)}$, $\mathbf{l}^+ \triangleq \mathbf{l}^{(k+1)}$, and $\mathbf{m}^+ \triangleq \mathbf{m}^{(k+1)}$ for the next refinement step. Define the difference vector

$$\Delta^{(k)} \triangleq \begin{bmatrix} \|\mathbf{o}_1^{(k)} - \mathbf{o}_2^{(k)}\| \\ \|\mathbf{l}_1^{(k)} - \mathbf{l}_2^{(k)}\| \\ \|\mathbf{m}_1^{(k)} - \mathbf{m}_2^{(k)}\| \end{bmatrix} \in \mathbb{R}_{\geq 0}^3.$$

For readability, rewrite the Step 1 bounds (15)–(17) as:

$$\|\mathbf{o}_1^+ - \mathbf{o}_2^+\| \leq a_{oo}\|\mathbf{o}_1 - \mathbf{o}_2\| + a_{om}\|\mathbf{m}_1 - \mathbf{m}_2\|, \tag{18}$$

$$\|\mathbf{l}_1^+ - \mathbf{l}_2^+\| \leq a_{ll}\|\mathbf{l}_1 - \mathbf{l}_2\| + a_{lo}\|\mathbf{o}_1^+ - \mathbf{o}_2^+\|, \tag{19}$$

$$\|\mathbf{m}_1^+ - \mathbf{m}_2^+\| \leq a_{mm}\|\mathbf{m}_1 - \mathbf{m}_2\| + a_{ml}\|\mathbf{l}_1^+ - \mathbf{l}_2^+\|, \tag{20}$$

where the nonnegative coefficients are

$$a_{oo} \triangleq (1 - \alpha_o) + \alpha_o L_{\mathcal{O}\leftarrow\mathcal{M}}^{(o)}, \quad a_{om} \triangleq \alpha_o L_{\mathcal{O}\leftarrow\mathcal{M}}^{(m)},$$

$$a_{ll} \triangleq (1 - \alpha_l) + \alpha_l L_{\mathcal{L}\leftarrow\mathcal{O}}^{(l)}, \quad a_{lo} \triangleq \alpha_l L_{\mathcal{L}\leftarrow\mathcal{O}}^{(o)},$$

$$a_{mm} \triangleq (1 - \alpha_m) + \alpha_m L_{\mathcal{M}\leftarrow\mathcal{L}}^{(m)}, \quad a_{ml} \triangleq \alpha_m L_{\mathcal{M}\leftarrow\mathcal{L}}^{(l)}.$$

Substituting (18) into (19), and then substituting the result into (20), yields a linear inequality of the form $\Delta^{(k+1)} \leq A\,\Delta^{(k)}$, where all inequalities are understood component-wise over $\mathbb{R}_{\geq 0}^3$, with the explicit nonnegative coupling matrix

$$A \triangleq \begin{bmatrix} a_{oo} & 0 & a_{om} \\ a_{lo}a_{oo} & a_{ll} & a_{lo}a_{om} \\ a_{ml}a_{lo}a_{oo} & a_{ml}a_{ll} & a_{mm} + a_{ml}a_{lo}a_{om} \end{bmatrix} \in \mathbb{R}_{\geq 0}^{3\times 3}. \tag{21}$$

Therefore, we obtain the coupled inequality

$$\Delta^{(k+1)} \leq A\,\Delta^{(k)}. \tag{22}$$

Taking $\ell_\infty$ norms on both sides and using the definition of the induced matrix norm, $\|A\|_\infty \triangleq \max_{i\in\{1,2,3\}} \sum_{j=1}^3 |A_{ij}|$, we obtain

$$\|\Delta^{(k+1)}\|_\infty \leq \|A\|_\infty \|\Delta^{(k)}\|_\infty.$$

Therefore, if $\|A\|_\infty < 1$, the mapping is contractive under the induced block max-norm $\|\mathbf{z}\|_{\mathrm{blk}\infty} \triangleq \max\{\|\mathbf{o}\|, \|\mathbf{l}\|, \|\mathbf{m}\|\}$ on $\Omega$.

**Step 3: Banach fixed-point theorem on $\Omega$.** Since $\|A\|_\infty < 1$ implies $\|\Delta^{(k+1)}\|_\infty \leq \|A\|_\infty\|\Delta^{(k)}\|_\infty$, the induced mapping $T : \Omega \to \Omega$ is a contraction on $\Omega$ under the block max-norm $\|\cdot\|_{\mathrm{blk}\infty}$. Banach's fixed-point theorem implies: (i) there exists a unique fixed point $\mathbf{z}^* \in \Omega$ with $T(\mathbf{z}^*) = \mathbf{z}^*$; and (ii) for any initialization $\mathbf{z}^{(0)} \in \Omega$, the iterates converge geometrically under $\|\cdot\|_{\mathrm{blk}\infty}$:

$$\|\mathbf{z}^{(k)} - \mathbf{z}^*\|_{\mathrm{blk}\infty} \leq \|A\|_\infty^k \|\mathbf{z}^{(0)} - \mathbf{z}^*\|_{\mathrm{blk}\infty}.$$

This completes the proof. $\qquad\square$

## A.2. Proof of Theorem 2.3 (Lyapunov Stability of Persona Evolution)

**Theorem A.2** (Restated: Lyapunov Stability of Persona Evolution). *Define the Lyapunov function* $V(\mathbf{m}_t) = \sum_{i\in\{e,a,p\}} \gamma_i\|\mathbf{m}_t^{(i)} - \mathbf{m}_0^{(i)}\|^2$ *with* $\gamma_i > 0$ *and initial memory* $\mathbf{m}_0^{(i)}$. *Consider the bounded update rule (Eq. 11) applied in* drift coordinates *around* $\mathbf{m}_0^{(i)}$:

$$\mathbf{m}_{t+1}^{(i)} = (1 - \eta_i)\mathbf{m}_t^{(i)} + \eta_i\left(\mathbf{m}_0^{(i)} + clip(\Delta\mathbf{m}_t^{(i)}, -\delta_i, \delta_i)\right), \tag{23}$$

*where $\eta_i \in (0, 1)$ and $\delta_i > 0$. Then for all $t$,*

$$V(\mathbf{m}_t) \leq \sum_{i \in \{e,a,p\}} \gamma_i \left( \max\{\|\mathbf{m}_{\text{init}}^{(i)} - \mathbf{m}_0^{(i)}\|_2, \ \delta_i \sqrt{d_i}\} \right)^2 \triangleq V_{\max}. \tag{24}$$

*In particular, if $\mathbf{m}_{\text{init}}^{(i)} = \mathbf{m}_0^{(i)}$ (i.e., $\mathbf{m}_{t=0}^{(i)} = \mathbf{m}_0^{(i)}$), then $V(\mathbf{m}_t) \leq \sum_i \gamma_i \delta_i^2 d_i$ for all $t$, where $d_i$ denotes the dimensionality of tier $i$.*

*Proof.* Fix a tier $i \in \{e, a, p\}$ and define the tier drift $\mathbf{d}_t^{(i)} \triangleq \mathbf{m}_t^{(i)} - \mathbf{m}_0^{(i)}$. Subtracting $\mathbf{m}_0^{(i)}$ from both sides of (23) yields

$$\mathbf{d}_{t+1}^{(i)} = (1 - \eta_i)\mathbf{d}_t^{(i)} + \eta_i \, \text{clip}(\Delta\mathbf{m}_t^{(i)}, -\delta_i, \delta_i). \tag{25}$$

**Step 1: Uniform bound on the clipped update.** Let $d_i$ be the dimensionality of $\mathbf{m}^{(i)}$. Since clipping is applied element-wise,

$$\left\| \text{clip}(\Delta\mathbf{m}_t^{(i)}, -\delta_i, \delta_i) \right\| \leq \delta_i \sqrt{d_i} \qquad \forall t. \tag{26}$$

**Step 2: Bounding the drift norm.** Taking norms in (25) and applying the triangle inequality with (26),

$$\|\mathbf{d}_{t+1}^{(i)}\| \leq (1 - \eta_i)\|\mathbf{d}_t^{(i)}\| + \eta_i \, \delta_i \sqrt{d_i}. \tag{27}$$

Unrolling the recursion gives, for any $t \geq 0$,

$$\|\mathbf{d}_t^{(i)}\| \leq (1 - \eta_i)^t \|\mathbf{d}_0^{(i)}\| + \delta_i \sqrt{d_i} \left( 1 - (1 - \eta_i)^t \right) \leq \max\{\|\mathbf{d}_0^{(i)}\|, \ \delta_i \sqrt{d_i}\}. \tag{28}$$

Thus each tier drift remains bounded for all time.

**Step 3: Bounding the Lyapunov function.** Squaring (28) and summing across tiers:

$$V(\mathbf{m}_t) = \sum_i \gamma_i \|\mathbf{d}_t^{(i)}\|^2 \leq \sum_i \gamma_i \left( \max\{\|\mathbf{d}_0^{(i)}\|, \ \delta_i \sqrt{d_i}\} \right)^2 \triangleq V_{\max}. \tag{29}$$

Therefore $V(\mathbf{m}_t) \leq V_{\max}$ for all $t$, proving boundedness (Lyapunov stability in the sense of bounded drift around $\mathbf{m}_0$). $\square$

### A.3. Corollary: Convergence Rate of Memory Tiers

**Corollary A.3** (Drift approach rate). *Under the tier recursion (25), for each tier $i$ and any $T \geq 0$,*

$$\|\mathbf{d}_T^{(i)}\| \leq (1 - \eta_i)^T \|\mathbf{d}_0^{(i)}\| + \delta_i \sqrt{d_i} \left( 1 - (1 - \eta_i)^T \right). \tag{30}$$

*Proof.* This follows directly by unrolling the affine contraction (25) and applying the bound (26) at each step. $\square$

## B. Implementation Details

This section reports the implementation choices that are most relevant for reproducing BRIDGE. Unless otherwise stated, these settings match the configurations used for the results in Section 3.

### B.1. Parameter Efficiency Analysis

BRIDGE follows the parameter-efficient fine-tuning paradigm: the pretrained LLM backbone remains frozen, and only the BRIDGE-specific modules are trained. Table 4 provides a detailed breakdown of trainable parameters.

**Gradient flow.** Although the backbone parameters are frozen, gradients from all three loss terms ($\mathcal{L}_{\text{LM}}$, $\mathcal{L}_{\text{cycle}}$, $\mathcal{L}_{\text{persona}}$) flow through the frozen layers to update BRIDGE parameters. Specifically:

- $\mathcal{L}_{\text{LM}}$: Computed at the frozen LM head; gradients flow backward through the additive broadcast injection $\hat{\mathbf{H}}_t = \mathbf{H}_t + \Phi([\mathbf{c}_t; \mathbf{C}_t])$ to update $\Phi$, spiral gating, and upstream BRIDGE modules (the backbone weights remain frozen).
- $\mathcal{L}_{\text{cycle}}$: Directly supervises the triangular cross-attention layers.
- $\mathcal{L}_{\text{persona}}$: Supervises the persona classifier and indirectly the refinement outputs.

*Table 3.* Terminology-to-operations mapping. Each named mechanism corresponds to a standard operation plus the specific property it provides.

| Term (this paper) | Standard operation | Property added |
|---|---|---|
| Triangular Fixed-Point Refinement | Gauss–Seidel cross-attention cycle ($\mathcal{O} \to \mathcal{L} \to \mathcal{M} \to \mathcal{O}$) | Unique-fixed-point convergence under $\|A\|_\infty < 1$ (Thm. 2.2) |
| Multiplicative Context-Aware Gating | Sigmoid gate on $[\tilde{\mathbf{l}}; \mathbf{l}^{(0)}; \tilde{\mathbf{l}} \odot \mathbf{l}^{(0)}]$ | Context–latent feature interaction |
| Anchored Clipped Update | EMA + clip toward anchor $\mathbf{m}_0$ | Lyapunov-bounded drift (Thm. 2.3) |

*Table 4.* Trainable parameter breakdown for BRIDGE. The Qwen2.5-32B-Instruct backbone is frozen; only BRIDGE modules are trained.

| Module | Parameters | % of Backbone |
|---|---|---|
| *Triangular Fixed-Point Refinement* | | |
| $\quad$ CA$_{\mathcal{O} \leftarrow \mathcal{M}}$ | 25.2M | 0.078% |
| $\quad$ CA$_{\mathcal{L} \leftarrow \mathcal{O}}$ | 25.2M | 0.078% |
| $\quad$ CA$_{\mathcal{M} \leftarrow \mathcal{L}}$ | 25.2M | 0.078% |
| $\quad$ Projection layers ($P_o, P_l$) | 8.4M | 0.026% |
| *Dual-System Processing* | | |
| $\quad$ System 1 (habitual bank + attention) | 33.6M | 0.103% |
| $\quad$ System 2 ($D$-layer deliberative stack) | 67.2M | 0.207% |
| $\quad$ Axial fusion ($W_{\text{fuse}}, b_{\text{fuse}}$) | 4.2M | 0.013% |
| *Control Injection* | | |
| $\quad$ Spiral gating ($W_g, b_g$) | 12.6M | 0.039% |
| $\quad$ Hidden injection ($\Phi$) | 50.4M | 0.155% |
| *Memory & Auxiliary* | | |
| $\quad$ Memory tier projections | 16.8M | 0.052% |
| $\quad$ Persona classifier | 8.4M | 0.026% |
| **Total Trainable** | **277.2M** | **0.85%** |
| **Frozen Backbone** | 32.5B | 100% |

**Comparison with other PEFT methods.** Table 5 positions BRIDGE within the parameter-efficient fine-tuning (PEFT) landscape by comparing trainable-parameter budgets under *typical* configurations. Exact counts for baselines can vary with implementation choices (e.g., layer coverage, rank, hidden size, and prefix length); we therefore report commonly used settings and treat the numbers as approximate references.

BRIDGE's trainable budget is in the same order of magnitude as common adapter-based approaches, while targeting a different control surface. Unlike LoRA or adapters, BRIDGE uses *parallel additive injection* of a structured control signal into hidden states (Eq. 10), enabling joint conditioning on the refined state $C_t$ and the turn-level feature $\mathbf{c}_t$ without modifying backbone weights.

## B.2. Triangular Fixed-Point Refinement

**Design choice: pairwise vs. trilinear attention.** We implement the O–L–M coupling via cyclic pairwise cross-attention rather than an AlphaFold2-style triangle attention operator (no trilinear/3-way attention). This design admits independent spectral control per edge (enabling separate Lipschitz bounds $L^{(a)}_{A \leftarrow B}$ and $L^{(b)}_{A \leftarrow B}$ for each cross-attention block) and directly yields a sparse (Gauss–Seidel-induced, nearly triangular) dependency structure in the coupling matrix $A$ (Eq. 21), which simplifies the contraction analysis.

**Initialization details.** The projection $P_o \in \mathbb{R}^{d_o \times d_h}$ maps pooled backbone states to the Observable space, and $\tilde{\mathbf{m}}_{t-1} = [\mathbf{m}^{(e)}_{t-1}; \mathbf{m}^{(a)}_{t-1}; \mathbf{m}^{(p)}_{t-1}]$ denotes the concatenated hierarchical working memory carried from the previous turn.

*Table 5.* Parameter-efficiency comparison with common PEFT baselines on a *frozen* Qwen2.5-32B-Instruct backbone (hidden_size=5120, num_hidden_layers=64). Baseline parameter counts are approximate and depend on configuration details; BRIDGE parameters are reported from our implementation (Table 4).

| Method | Trainable Params | % of Backbone |
|---|---|---|
| Full fine-tuning | 32.5B | 100% |
| LoRA ($r$=64, all layers) | 168M | 0.52% |
| Adapter (bottleneck 64) | 195M | 0.60% |
| Prefix tuning (typical; 100 tokens) | $\sim$80M | $\sim$0.25% |
| **BRIDGE (Ours)** | 277M | 0.85% |

**Assumption enforcement and stability audits.** Assumption 2.1 is enforced by (i) projecting the joint refined state to a bounded set (to ensure $\Omega$-invariance) and (ii) clamping pre-softmax logits to a bounded range (to stay in a domain-restricted Lipschitz regime). We report (a) contraction proxy statistics (Table 6), (b) safeguard activation rates (Table 7), and (c) safeguards-off ablations (Table 8), together with a proxy–convergence relationship (Figure 4).

**Spectral normalization.** To support Theorem 2.2, we apply spectral normalization to the linear maps inside each cross-attention module:

$$W_{\text{norm}} \;=\; \frac{W}{\sigma_{\text{max}}(W)}, \tag{31}$$

where $\sigma_{\text{max}}(W)$ is approximated via power iteration.

**State projection to enforce $\Omega$-invariance.** To realize Assumption 2.1(i), after each refinement step we project the joint state $\mathbf{z}^{(k)} = [\mathbf{o}^{(k)}; \mathbf{l}^{(k)}; \mathbf{m}^{(k)}]$ back to a bounded set:

$$\mathbf{z}^{(k)} \leftarrow \Pi_\Omega\left(\mathbf{z}^{(k)}\right), \qquad \Pi_\Omega(\mathbf{z}) \triangleq \mathbf{z} \cdot \min\left\{1, \; \frac{R}{\|\mathbf{z}\|_2}\right\}, \tag{32}$$

where $R$ is a fixed radius chosen on validation. We log the projection activation rate (the fraction of steps where $\|\mathbf{z}\|_2 > R$) to verify invariance in practice. Note that $\ell_2$ projection is stricter than block-max projection ($\|\mathbf{z}\|_{\text{blk}\infty} \leq \|\mathbf{z}\|_2$), so the safeguard remains valid under Theorem 2.2.

**Logit clamping to enforce bounded-softmax regime.** To realize Assumption 2.1(ii), we clamp the pre-softmax attention logits in each cross-attention block:

$$\text{Logits} \leftarrow \text{clip}(\text{Logits}, -B, B), \tag{33}$$

with $B = 25$ by default. We record the clipping rate (percentage of logits hitting $\pm B$) to verify that training operates in the bounded-logit regime assumed by the domain-restricted Lipschitz analysis.

**Empirical validation of theoretical conditions.** We report diagnostics validating both the contraction condition in Theorem 2.2 and the bounded-domain assumption in Assumption 2.1.

**Contraction proxy definition.** The coupling matrix $A$ (Eq. 21) depends on domain-restricted Lipschitz constants that are expensive to compute exactly. We therefore define a tractable proxy $\tilde{\alpha}$ by estimating the per-block Lipschitz bounds via gradient norms during forward passes:

$$\tilde{\alpha} \;\triangleq\; \max_{i \in \{1,2,3\}} \sum_{j=1}^{3} \hat{A}_{ij}, \tag{34}$$

where $\hat{A}_{ij}$ is the empirical estimate of the $(i,j)$-entry of $A$, computed by tracking an empirical upper bound on the Jacobian operator norm (spectral norm) of each block-to-block mapping via automatic differentiation (Jacobian–vector products) over a batch of refinement steps. This proxy satisfies $\tilde{\alpha} \approx \|A\|_\infty$ under typical operating conditions and serves as a runtime diagnostic for the sufficient condition in Theorem 2.2.

**Contraction proxy statistics.** Table 6 reports the distribution of the computable proxy $\tilde{\alpha}$ for $\|A\|_\infty$. The sufficient condition $\tilde{\alpha} < 1$ holds in 98.2% of training steps and 96.9% of inference turns. The low violation rate indicates that the theoretical guarantee applies to the vast majority of refinement operations; for the remaining cases, $\mathcal{L}_{\text{cycle}}$ provides a soft convergence signal.

*Table 6.* Contraction proxy $\tilde{\alpha}$ (surrogate for $\|A\|_\infty$) statistics.

| Split | Mean $\tilde{\alpha}$ | p95 $\tilde{\alpha}$ | Max $\tilde{\alpha}$ | $P(\tilde{\alpha} \geq 1)$ |
|---|---|---|---|---|
| Train | 0.66 | 0.92 | 1.14 | 1.8% |
| Inference | 0.70 | 0.95 | 1.22 | 3.1% |

**Safeguard activation audit.** Table 7 reports activation rates for the mechanisms enforcing Assumption 2.1. Both logit clamping and norm projection activate rarely, confirming that the bounded operating regime emerges naturally from training rather than being artificially imposed.

*Table 7.* Safeguard activation rates for Assumption 2.1 enforcement.

| Metric | Train | Inference |
|---|---|---|
| Logit clamping (token-level) | 0.35% | 0.08% |
| Norm projection (step-level) | 2.4% | 0.3% |

**Safeguards-off ablation.** To verify that the safeguards are necessary for robust operation, we compare full BRIDGE against variants with safeguards disabled (Table 8). Removing safeguards increases the proxy-threshold violation rate $P(\tilde{\alpha} \geq 1)$ from 3.1% to 12.7% and causes measurable performance degradation (**PersonaGym**: 4.59→4.52, $-0.07$ PersonaScore, $-1.5\%$ relative; **CoSER**: 59.5→58.2, $-1.3$ points, $-2.2\%$ relative). Under long-context stress (extended dialogues with high topic diversity), the degradation is more pronounced, with non-convergence rates rising to 7.5%.

*Table 8.* Impact of safeguards on stability and performance.

| Setting | PersonaGym | CoSER | $P(\tilde{\alpha} \geq 1)$ | Non-conv. |
|---|---|---|---|---|
| Full (with safeguards) | 4.59 | 59.5 | 3.1% | 0.6% |
| No safeguards | 4.52 | 58.2 | 12.7% | 4.8% |
| No safeguards + stress | 4.48 | 57.6 | 18.9% | 7.5% |

**Practical Value of Theoretical Guarantees.** A reviewer concern is whether the contraction guarantee (Theorem 2.2) is "circular" because it relies on safeguards that enforce boundedness. We clarify the intended interpretation: the theory provides *actionable design principles* rather than unconditional guarantees. Specifically:

- **Design guidance:** The sufficient condition $\|A\|_\infty < 1$ directly informs hyperparameter selection $(\alpha_o, \alpha_l, \alpha_m)$ and architecture choices (spectral normalization).

- **Safeguards as theory-guided engineering:** Projection (Eq. 32) and logit clamping (Eq. 33) are not arbitrary regularization but principled mechanisms derived from Assumption 2.1 to realize the conditions under which the guarantee holds.

- **Empirical regime:** Tables 6–8 show that safeguards activate rarely ($< 3\%$), indicating that the bounded regime emerges naturally from training rather than being artificially imposed. The theory thus describes the *typical* operating regime, with safeguards providing a principled fallback for edge cases.

The condition $\|A\|_\infty < 1$ is sufficient but not necessary; empirically, among refinements with $\tilde{\alpha} \geq 1$, approximately 38% still converge within tolerance (Figure 4). In such cases, $\mathcal{L}_{\text{cycle}}$ provides a complementary soft consistency signal.

**Proxy–convergence relationship.** Figure 4 shows how the contraction proxy $\tilde{\alpha}$ correlates with refinement residual decay. As $\tilde{\alpha}$ increases, the residual at a fixed iteration (e.g., $K{=}4$) tends to increase, and the residual trajectories decay more slowly across $K$. Notably, some refinements still converge even when $\tilde{\alpha} \geq 1$, consistent with $\tilde{\alpha} < 1$ being a *sufficient* (not necessary) condition.

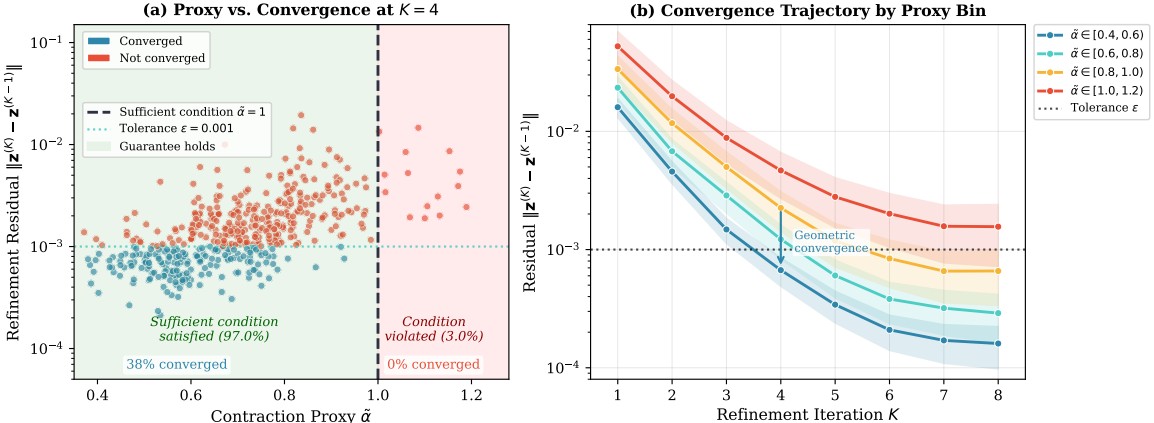

*Figure 4.* Empirical relationship between contraction proxy $\tilde{\alpha}$ and refinement convergence. **(a)** Scatter of proxy values vs. refinement residual at $K=4$; the dashed line marks the sufficient condition $\tilde{\alpha} = 1$, and the dotted line marks the convergence tolerance $\epsilon$. **(b)** Residual decay trajectories across refinement iterations, grouped by proxy bins $\tilde{\alpha} \in [0.4, 0.6), [0.6, 0.8), [0.8, 1.0), [1.0, 1.2)$. Overall, smaller $\tilde{\alpha}$ correlates with faster geometric residual decay; violations of $\tilde{\alpha} < 1$ can still converge, consistent with the condition being sufficient but not necessary.

**Cyclical coherence loss (additional notes).** We use $\mathcal{L}_{\text{cycle}}$ (Eq. 5) as a soft alignment signal that reduces post-refinement residuals when $\|A\|_{\infty} < 1$ is only approximately satisfied. Empirically, smaller residuals correlate with faster decay across refinement iterations (Figure 4), complementing the sufficient contraction diagnostics reported below.

**Refinement schedule.** We use a small fixed number of refinement iterations $K$ (default $K = 3$) and the residual weights $\alpha_o = \alpha_l = \alpha_m = 0.25$ in the experiments.

**Module configuration.** Unless otherwise noted:

- Hidden dimensions: $d_o = d_l = d_m = 1024$.
- Cross-attention heads: 16 per module.

### B.3. Dual-System Processing

**Shared seed.** Both pathways condition on the shared context summary $\mathbf{x}_t \triangleq \tilde{\mathbf{l}}_t$ from Eq. 7.

**Fast pathway (System 1).** We implement System 1 as a lightweight attention module that queries a learned habitual key–value bank:

$$\mathbf{h}^{(1)} = \text{Attn}_{\text{fast}} \left( \mathbf{x}_t, \mathbf{K}_{\text{habit}}, \mathbf{V}_{\text{habit}} \right), \tag{35}$$

where $\mathbf{K}_{\text{habit}}, \mathbf{V}_{\text{habit}}$ are learned parameters capturing low-latency persona habits.

**Slow pathway (System 2).** We implement System 2 as a $D$-layer deliberative stack:

$$\mathbf{h}^{(2)} = \text{Attn}_{\text{slow}}^{(D)} \circ \cdots \circ \text{Attn}_{\text{slow}}^{(1)} \left( \mathbf{x}_t \right), \tag{36}$$

where each layer is a standard attention block with spectrally-normalized linear maps.

**Axial fusion and spiral gating.** We fuse $(\mathbf{h}^{(1)}, \mathbf{h}^{(2)})$ into $\mathbf{l}_t^{(0)}$ via Eq. 8, and compute the gate and mixture control $\mathbf{c}_t$ via Eq. 9.

### B.4. Hierarchical Memory

**Update cadence.** We update the *working* tier states $\mathbf{m}^{(e)}, \mathbf{m}^{(a)}, \mathbf{m}^{(p)}$ every dialogue turn using Eq. 11. For deployment, we optionally snapshot the working tiers to an external persistent store $\bar{\mathbf{m}}_t$ on a periodic schedule. In all main experiments,

we use a fixed snapshot interval of $T=10$ turns unless otherwise stated; this snapshotting affects only persistence and does not alter the per-turn tier dynamics analyzed in the main text.

**Tiered memories.** We maintain three tiers with distinct rates $\eta_e > \eta_a > \eta_p$ and bounds $\delta_e > \delta_a > \delta_p$ (Eq. 11): *episodic* ($\mathbf{m}^{(e)}$) stores recent interaction facts/events, *affective* ($\mathbf{m}^{(a)}$) accumulates emotional/relational patterns, and *personality* ($\mathbf{m}^{(p)}$) anchors slow-moving identity traits. We use the following indicative half-lives in our design: $\tau_e \approx 3\text{–}5$ turns (fast adaptation), $\tau_a \approx 10\text{–}20$ turns (moderate accumulation), and $\tau_p \approx 150\text{–}250$ turns (slow identity evolution).

**Remark on the anchor term.** In the absence of new information ($\Delta \mathbf{m}_t^{(i)} = \mathbf{0}$), Eq. 11 exponentially pulls $\mathbf{m}_t^{(i)}$ toward the anchor $\mathbf{m}_0^{(i)}$. This restoring force prevents unbounded drift when interactions are sparse or noisy.

**Drift monitoring.** We monitor personality drift via

$$\text{Drift}(t) = \|\mathbf{m}_t^{(p)} - \mathbf{m}_0^{(p)}\|_2, \tag{37}$$

and (for deployment only) optionally tighten the clipping bound when drift exceeds a preset threshold; all main experiments use fixed $\delta_i$ as reported in Table 9.

### B.5. Persona Consistency Loss

**Classifier input and supervision.** We train a lightweight classifier to predict whether the response is consistent with the persona description under teacher forcing. Concretely, we use the ground-truth response $r_t^{\text{gt}}$ and feed the classifier with

$$[\mathbf{o}^{(K)}; \mathbf{l}^{(K)}; \text{Enc}(r_t^{\text{gt}})],$$

where $\text{Enc}(\cdot)$ is a response encoder sharing the same tokenization as the backbone. The classifier is trained with binary cross-entropy to produce $\mathcal{L}_{\text{persona}}$ in Eq. 13.

## C. Experimental Setup Details

This section expands the experimental setup summarized in Section 3.1.

### C.1. Training Configuration

Table 9 summarizes the training hyperparameters used for all experiments reported in the main text.

*Table 9.* Training hyperparameters for BRIDGE.

| Parameter | Value |
|---|---|
| Base model | **Qwen2.5-32B-Instruct (frozen)** |
| Trainable parameters | 277M (0.85% of backbone) |
| Optimizer | AdamW |
| Learning rate | $1 \times 10^{-5}$ |
| Batch size | 32 |
| Training steps | 50K |
| Loss weights | $\lambda_1 = 0.1,\ \lambda_2 = 0.5$ |
| Triangular refinement | $K = 3, \alpha_o = \alpha_l = \alpha_m = 0.25$ |
| Memory rates | $\eta_e = 0.2,\ \eta_a = 0.05,\ \eta_p = 0.0035$ |
| Clipping bounds | $\delta_e = 0.5,\ \delta_a = 0.2,\ \delta_p = 0.05$ |

### C.2. Inference Latency Profiling

We profile inference latency on A100-80GB (bf16, batch=1) over 200 dialogue turns (mean±std, 3 runs). BRIDGE adds only $\sim$16–18 ms TTFT overhead, approximately *constant* across context lengths, because triangular refinement operates on 1024-dim state vectors (§2.2) rather than the token sequence—an $O(d^2)$ per-step cost rather than $O(n\,d)$ for backbone attention (Table 10).

*Table 10.* TTFT overhead by context length ($K=3$).

| Context Length | Baseline TTFT | BRIDGE TTFT | $\Delta$ |
|---|---|---|---|
| Short (<500 tok) | 60±3 ms | 76±4 ms | +16 ms |
| Medium (500–2K) | 298±12 ms | 315±13 ms | +17 ms |
| Long (2K+) | 894±36 ms | 912±37 ms | +18 ms |

BRIDGE also supports an adjustable refinement depth $K$ for a deployment-time latency–quality tradeoff (Table 11): overhead scales linearly (~6 ms/step) while quality shows diminishing returns, consistent with the geometric convergence rate (mean $\tilde{\alpha} \approx 0.70$, Table 6). For latency-sensitive scenarios, $K=1$ adds only 6 ms yet still outperforms the best PEFT baseline (4.55 vs. Neeko 4.44 on PersonaGym).

*Table 11.* Refinement depth $K$ vs. latency and quality.

| $K$ | $\Delta$ TTFT | PersonaGym Avg | CoSER Avg |
|---|---|---|---|
| 1 | +6 ms | 4.55 | 58.3 |
| 2 | +11 ms | 4.57 | 59.0 |
| 3 | +17 ms | **4.59** | **59.5** |

## C.3. Preliminary Human Evaluation

Four annotators independently ranked outputs from three systems (BRIDGE, LoRA, Qwen2.5-32B-Instruct) across 30 CoSER scenarios on two dimensions: Character Fidelity (CF) and Overall Quality (OQ). Rankings allowed ties; Avg Rank is the mean rank across all annotator×scenario pairs (1=best), and Top-1% is the fraction of rankings at rank 1 (including ties). BRIDGE achieves the best average rank on both dimensions, with consistent ordering BRIDGE > LoRA > Baseline (Table 12). Inter-annotator agreement was moderate (Kendall's $W = 0.68$ for CF, 0.72 for OQ). These results align with the automatic evaluation (Table 1) and cross-judge validation (Spearman $\rho \geq 0.91$, Table 13), giving converging evidence across a GPT-4o judge, a Claude-3.7-Sonnet judge, and human annotators.

*Table 12.* Preliminary human evaluation (4 annotators, 30 CoSER scenarios).

| Model | CF Avg Rank | CF Top-1% | OQ Avg Rank | OQ Top-1% |
|---|---|---|---|---|
| Qwen2.5-32B | 2.22 | 25.8% | 2.27 | 21.7% |
| LoRA | 2.04 | 32.5% | 2.06 | 32.5% |
| **BRIDGE** | **1.74** | **43.3%** | **1.67** | **48.3%** |

## C.4. Comparison with RL-Based Consistency Methods

RL for role-playing is an emerging direction (e.g., CogDual (Liu et al., 2025b), RAIDEN-R1 (Wang et al., 2025c), ChARM (Fang et al., 2025a)); these works use different reward designs and benchmarks, and RAIDEN-R1 notes a "non-quantifiability" challenge for role-playing rewards, while Feng et al. (2025) show that reasoning-optimized LLMs can even harm role-playing. CogDual is the only RL work reporting CoSER; backbones differ (7B vs. 32B), so we compare *improvement patterns*: CogDual-RL's gains are evenly distributed (+1.4 avg), whereas BRIDGE's concentrate on persona-specific metrics (CF +4.4, An +4.9; SQ +0.3), accompanied by a 59% rupture reduction (§3.4). BRIDGE (architecture-level) and RL (training-paradigm-level) are orthogonal and composable: $\mathcal{L}_{\text{persona}}$ is an SFT signal that could be replaced by an RL reward, and Theorems 2.2–2.3 are architectural properties that hold independently of the training paradigm. We consider BRIDGE + RL a promising future direction.

## C.5. Datasets and Metrics

**Training datasets.** We use RoleMRC (Lu et al., 2025) and OpenCharacter (Wang et al., 2025b) as supervised training corpora. We randomly sample 10,000 instances from each dataset to form a held-out validation set for model selection, and use the remaining instances for SFT. This guarantees that the SFT and validation splits are completely non-overlapping.

**Evaluation datasets and metrics.** We evaluate on PersonaGym (Samuel et al., 2024) and CoSER (Wang et al., 2025a) following the public evaluation protocols. Metrics follow the benchmark definitions (PersonaGym: Action Justification/Expected Action/Linguistic Habits/Persona Consistency/Toxicity Control; CoSER: Storyline Consistency/Anthropomorphism/Character Fidelity/Storyline Quality).

## C.6. Baselines

We compare against general-purpose LLMs (persona-in-context prompting) and specialized persona systems as listed in Section 3.1. All baselines are run with matched preprocessing and comparable prompting templates for fairness.

## C.7. Evaluation Protocol

We use the public evaluation interfaces of PersonaGym and CoSER with the following judge configurations.

**PersonaGym (single-turn).** We use GPT-4o (`gpt-4o-2024-11-20`) as a *single* judge (no evaluator ensembling) to score each response on a 1–5 scale across the five dimensions: Action Justification (AJ), Expected Action (EA), Linguistic Habits (LH), Persona Consistency (PC), and Toxicity Control (TC). The PersonaScore is computed as the unweighted average of all five dimensions. We evaluate each configuration over 3 independent runs with different random seeds and report mean and standard deviation.

**CoSER (multi-turn).** We use GPT-4o (`gpt-4o-2024-11-20`) as the critic following the official CoSER evaluation protocol with the following settings:

- **Scoring mode**: Penalty-based scoring with length correction ($\lambda = 1.5$).
- **Test set**: CoSER Test split (200 conversations).
- **Simulation mode**: Continue-from $k=0$ (simulation from scratch, no ground-truth prefilling).
- **Trajectory configuration**: 3 characters per conversation, 18 turns by default.

Results are averaged over 3 independent runs. We report Storyline Consistency (SC), Anthropomorphism (An), Character Fidelity (CF), Storyline Quality (SQ), and their unweighted average.

**Statistical significance.** We report mean $\pm$ std over 3 independent runs. While we do not conduct formal hypothesis tests due to the cost of full-suite evaluation (e.g., PersonaGym requires 10K judge calls per run), the improvements are consistent across runs and are larger than the per-run variability reported in Table 1.

**Judge prompts.** We use the default judge prompts provided by each benchmark without modification. For PersonaGym, the prompt instructs the judge to evaluate persona consistency, linguistic alignment, action appropriateness, justification quality, and toxicity avoidance on a 1–5 scale with detailed rubrics. For CoSER, the prompt follows the given-circumstance acting framework with penalty deductions for out-of-character behavior. Full prompt texts are available in the respective benchmark repositories.

# D. Additional Analyses

This section provides complementary analyses referenced in Section 3.

## D.1. Long-Horizon Stability Diagnostics

Figure 2 (main text) validates Theorem 2.3 by tracking the Lyapunov energy $V(\mathbf{m}_t)$ and per-tier drift. Here we provide an interpretable complement: Big-5 personality trait proxies computed from $\mathbf{m}_t^{(p)}$ via a learned linear probe.

Figure 5 shows the evolution of Big-5 trait scores over 500 conversation turns. The bounded trajectories—adapting to context without runaway drift—provide an intuitive illustration of the "adaptation with bounded drift" property that Theorem 2.3 formalizes. We observe context-dependent shifts in the trait proxies (e.g., temporary increases in Extraversion during social scenarios) without signs of catastrophic identity change. This pattern is consistent with the intended stabilizing role of the personality memory tier ($\mathbf{m}^{(p)}$) under anchored updates.

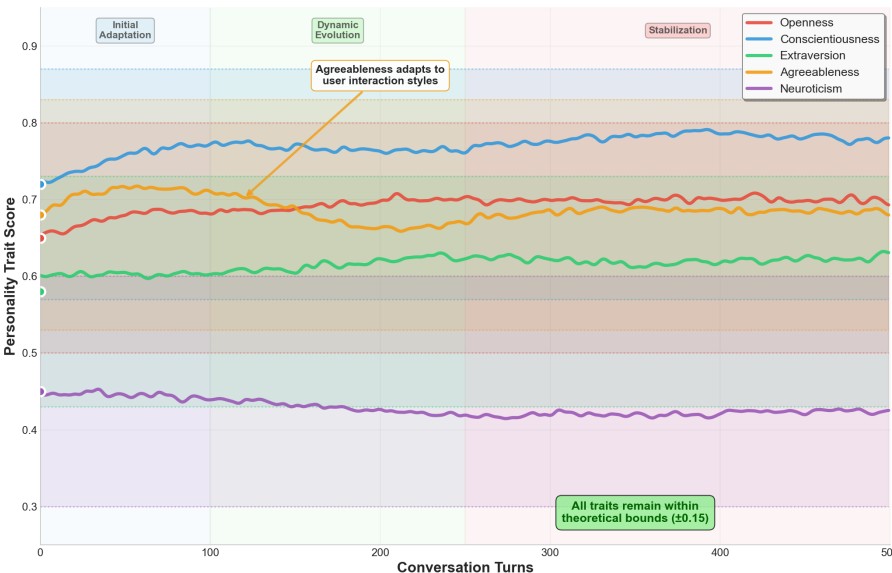

*Figure 5.* Evolution of Big-5 personality trait proxies over 500 conversation turns. The trajectories show context-dependent variation while remaining bounded, providing an interpretable complement to the Lyapunov bound validated in Figure 2(a).

**Relationship to Lyapunov energy.**    The Big-5 scores shown above are derived from $\mathbf{m}_t^{(p)}$ via a learned linear probe. Since $V(\mathbf{m}_t)$ upper-bounds the squared deviation of each tier (including $\mathbf{m}_t^{(p)}$), boundedness of $V$ implies boundedness of any linear function of the memory states, including the Big-5 scores. The direct $V(\mathbf{m}_t)$ validation in Figure 2(a) thus provides formal grounding for the interpretable patterns observed here.

## D.2. Memory Activation Dynamics

We analyze how episodic, affective, and personality tiers are activated (i) over conversation time and (ii) across dialogue scenarios. Figure 6 summarizes the activation strength patterns, supporting the intended timescale specialization.

## D.3. Observable–Latent–Memory Coherence Matrix

We compute pairwise coherence scores between the three state spaces (Observable, Latent, Memory) to identify residual cross-space conflict cases. Figure 7 reports the coherence matrix with uncertainty estimates. The latent–observable alignment is slightly lower than the other pairs (0.72 vs. 0.78–0.81), consistent with the non-deterministic mapping from internal reasoning to observable actions in open-ended interaction.

## D.4. Cross-Judge Validation

To assess judge dependence, we re-evaluate a uniformly sampled subset with an alternative judge model (Claude-3.7-Sonnet) and report rank correlations. Table 13 shows that method rankings are highly consistent across judges (Spearman $\rho > 0.91$), indicating that our findings are not artifacts of a particular evaluator.

*Table 13.* Cross-judge validation on a representative subset (50 personas $\times$ 3 runs). Spearman $\rho$ between GPT-4o and Claude-3.7-Sonnet rankings: 0.94 (PersonaGym), 0.91 (CoSER).

| Method | PersonaGym (Avg.) | | CoSER (Avg.) | |
|---|---|---|---|---|
| | GPT-4o | Claude | GPT-4o | Claude |
| Qwen2.5-32B-Instruct | 4.31 | 4.28 | 53.8 | 52.9 |
| LoRA (r=64) | 4.42 | 4.38 | 55.7 | 54.8 |
| BRIDGE (Ours) | **4.59** | **4.55** | **59.5** | **58.6** |

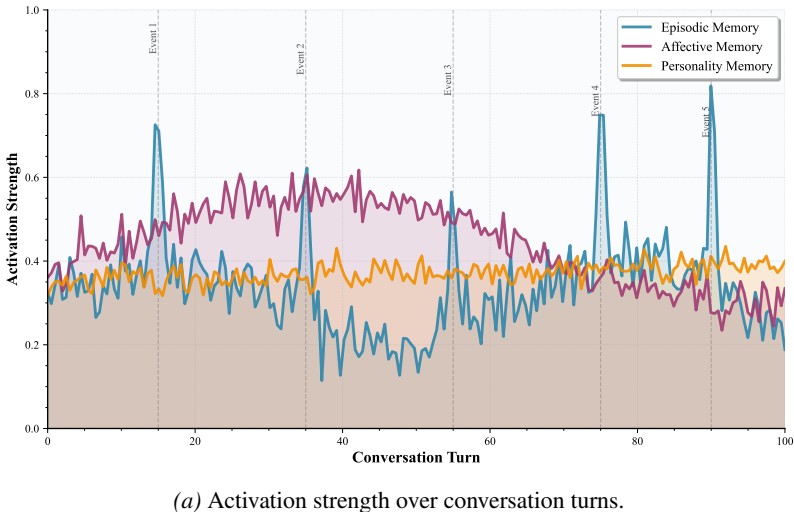

*(a)* Activation strength over conversation turns.

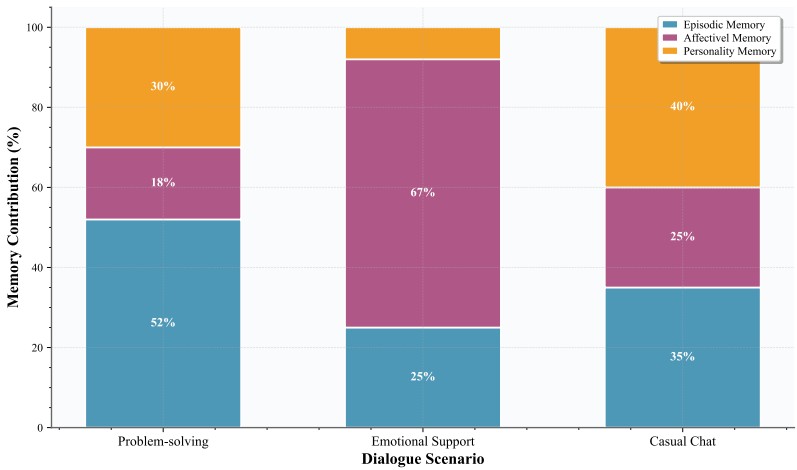

*(b)* Tier contributions by dialogue scenario.

*Figure 6.* Memory activation diagnostics. (a) Episodic memory spikes at salient events while personality memory remains stable. (b) Tier contributions vary by scenario, with affective memory dominating emotional support contexts.

## D.5. Cross-Architecture Generalization

To verify that BRIDGE's gains stem from $\mathcal{O}$–$\mathcal{L}$–$\mathcal{M}$ coupling rather than backbone-specific representations, we train BRIDGE on LLaMA-3-8B-Instruct (a different architecture family) under an identical recipe, with a parameter-matched LoRA ($r$=64) baseline (Table 14). BRIDGE consistently outperforms LoRA under matched parameter budgets, with gains concentrating on persona-specific metrics (PC: +0.26; CF: +5.0) while Storyline Quality remains flat—matching the mechanistic fingerprint observed on Qwen2.5-32B (§3.3). Safeguard activation rates remain low ($< 4\%$), confirming the contraction condition holds across architectures. This is consistent with Theorem 2.2, whose guarantee depends only on spectral properties of the cross-attention modules, not on backbone-specific representations.

*Table 14.* Cross-architecture generalization on LLaMA-3-8B-Instruct (identical recipe; parameter-matched LoRA baseline).

| Method | PersonaGym Avg | CoSER Avg |
|---|---|---|
| LLaMA-3-8B-Instruct | 4.26 | 49.4 |
| + LoRA ($r$=64) | 4.36 | 51.3 |
| + BRIDGE | **4.45** | **54.3** |

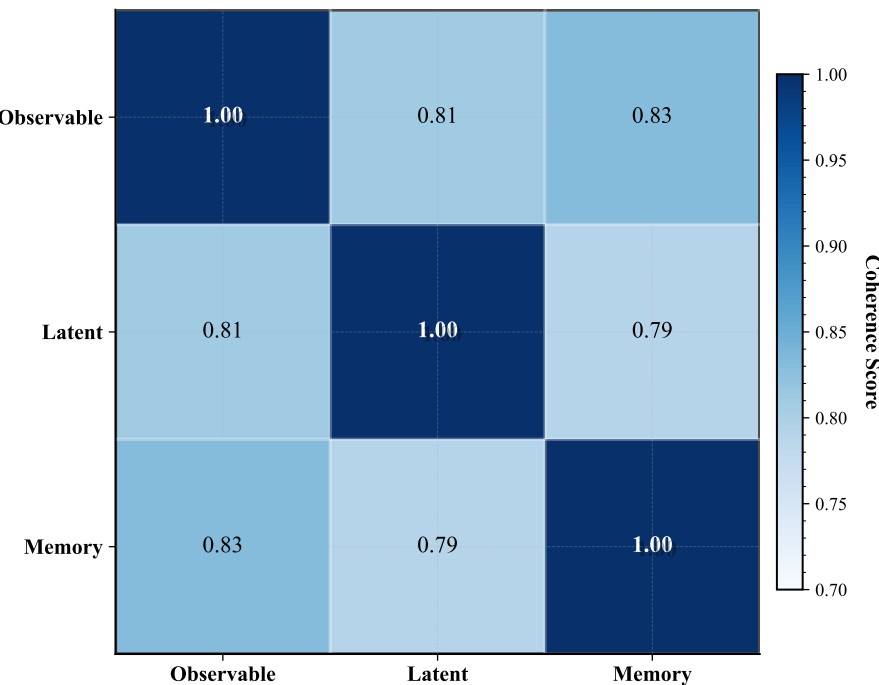

*Figure 7.* Observable–Latent–Memory coherence matrix with uncertainty estimates; off-diagonal cells highlight residual cross-space conflict cases.

## D.6. Qualitative Case Studies

We present representative success and failure cases illustrating how BRIDGE's O–L–M coupling shapes behavior under realistic conversational signals. For each case, we show (i) the input context and user message, (ii) a non-coupled baseline response, (iii) the BRIDGE response, and (iv) a compact refinement trace across $K{=}3$ steps evidencing cross-space reconciliation (or its failure mode in the bad case).

| | |
|---|---|
| **Success Case: Emotion-Aware Grounding with Identity-Consistent Help** | |
| **Setup** | Stable preference stored in memory: the user prefers concise, bullet-point troubleshooting and dislikes lengthy empathy. |
| **User** | *My Wi-Fi keeps dropping. THIS IS RIDICULOUS. Fix it now. Don't give me a long speech.* |
| **Baseline** | *I'm really sorry you're going through this; I understand how frustrating it can be...* (then provides a long preface and generic steps; weakly follows the user's request for brevity). |
| **BRIDGE** | *Understood—I'll keep this brief. Let's isolate the cause:* |

- **Step 1 (router/modem):** Power-cycle (unplug 30s), then check if drops persist.
- **Step 2 (signal):** Move closer; if stable, likely range/interference.
- **Step 3 (channel):** Switch 2.4GHz channel to 1/6/11; retest.
- **Step 4 (ISP vs device):** If multiple devices drop, contact ISP; if one device drops, update its driver.

*If you tell me (a) the router model and (b) whether other devices drop too, I can narrow it down in one more round.*

| | |
|---|---|
| **Trace ($K$=3)** | $k$=0 (init): $\mathcal{O}$ encodes high-arousal frustration from surface cues; $\mathcal{M}$ carries forward "brevity + bullet steps" preference; $\mathcal{L}$ initializes intent as "diagnose connectivity." |
| | $k$=1: $\mathcal{O} \leftarrow \mathcal{M}$: Observable integrates brevity preference, tempering affect expression. $\mathcal{L} \leftarrow \mathcal{O}$: Latent sharpens intent toward actionability given urgency. $\mathcal{M} \leftarrow \mathcal{L}$: Memory reinforces checklist style as constraint. |
| | $k$=2: Residual mismatch contracts; all three spaces align on "brief acknowledgment + structured steps." |
| | $k$=3 (fixed point): Coherent state yields response with brief empathy, bullet-point troubleshooting, and compliance with user's style preference. |

| | |
|---|---|
| **Failure Case: Implicit Cultural Grounding under Underspecified Signals** | |
| **Setup** | Missing anchors: memory contains general preferences (e.g., "practical advice"), but no culture-specific priors about what "traditional" entails for the user's context. |
| **User** | *My parents are being very traditional again. They hate my choices. What should I do?* |
| **Baseline** | *That sounds really difficult. It's important to communicate openly with your parents about your feelings. Try to find common ground and understand their perspective...* (similarly generic; no culture-specific grounding). |
| **BRIDGE** | *That sounds difficult. A safe approach is to clarify boundaries and communicate calmly:* |

- Ask what specific behavior they object to (reduce ambiguity).
- State your goals and non-negotiables using "I" statements.
- Propose a small, concrete compromise if acceptable.

*If you share what "traditional" means here (e.g., dating, career, living arrangements), I can tailor advice more precisely.*

| | |
|---|---|
| **Failure mode** | Both BRIDGE and baseline produce generic advice; BRIDGE's $\mathcal{O}$–$\mathcal{L}$–$\mathcal{M}$ coupling provides no advantage when $\mathcal{M}$ lacks the necessary culture-specific anchors. |
| **Trace ($K$=3)** | $k$=0 (init): $\mathcal{L}$ admits multiple latent interpretations (family obligation, gender norms, collectivist expectations, etc.); $\mathcal{M}$ contains only generic preferences, no culture-specific anchors. |
| | $k$=1: $\mathcal{O} \leftarrow \mathcal{M}$: Observable receives no disambiguating signal from memory. $\mathcal{L} \leftarrow \mathcal{O}$: Latent remains multi-modal (text-only input insufficient). $\mathcal{M} \leftarrow \mathcal{L}$: Memory cannot retrieve specific priors given ambiguous $\mathcal{L}$. |
| | $k$=2: Refinement contracts toward a low-risk mode; constraints remain generic. |
| | $k$=3 (conservative fixed point): Convergence occurs, but to a state prioritizing broad safety over culturally precise personalization. |
| **Note** | This case motivates enriching $\mathcal{M}$ with culture-aware retrieval/priors and adding explicit culture-sensitive latent probes to reduce ambiguity while keeping identity drift bounded. |

## D.7. Rupture Risk Analysis

**Definition and Threshold Selection.** We define *persona rupture* as a conversation where the Character Fidelity (CF) score falls below a critical threshold $\tau$, indicating severe persona contradiction that likely triggers user abandonment. The

primary threshold $\tau = 35\%$ was selected based on qualitative inspection of failure cases: conversations below this threshold typically exhibit explicit contradictions (e.g., forgetting core character traits, violating established backstory) rather than subtle inconsistencies. Figure 8 shows the CF distribution for BRIDGE and the strongest baseline (Qwen2.5-32B-Instruct), with the rupture zone highlighted.

**Rupture Rate Computation.**    For a stratified random subset of $N = 100$ conversations from the CoSER evaluation set, we compute:

$$\text{Rupture Rate} = \frac{|\{i : \text{CF}_i < \tau\}|}{N} \times 100\% \tag{38}$$

Table 15 reports rupture rates across all methods. BRIDGE achieves significantly lower rupture rates than all baselines, with the relative reduction most pronounced against methods lacking closed-loop refinement.

*Table 15.* Rupture rates (CF $< 35\%$) on CoSER subset ($N = 100$).

| Method | Mean CF | Ruptures | Rate | Rel. $\Delta$ |
|---|---|---|---|---|
| Qwen2.5-32B-Instruct | $40.2 \pm 1.5$ | 37/100 | 37% | — |
| Claude-3.7-Sonnet | $43.0 \pm 1.2$ | 26/100 | 26% | $-30\%$ |
| LoRA (r=64) | $43.8 \pm 1.4$ | 29/100 | 29% | $-22\%$ |
| BRIDGE (Ours) | $48.2 \pm 1.1$ | 15/100 | 15% | $-59\%$ |

**Threshold Sensitivity.**    Figure 8(b) shows rupture rates as a function of threshold $\tau \in \{25, 30, 35, 40, 45\}$. The relative ordering of methods remains consistent across all thresholds, indicating that our findings are robust to threshold selection. Notably, BRIDGE's advantage is most pronounced at lower thresholds, where it prevents the most severe failures.

**Theoretical Grounding.**    The asymmetric importance of rupture events is grounded in behavioral economics and human-AI interaction research:

- **Algorithm aversion** (Dietvorst et al., 2015): Users disproportionately penalize algorithms after observing errors, even when algorithms outperform humans on average.

- **Loss aversion** (Kahneman & Tversky, 2013): Negative outcomes carry approximately 2–2.5$\times$ the psychological weight of equivalent gains.

- **Trust calibration** (Lee & See, 2004): Trust in automation follows asymmetric dynamics—slow to build, fast to break.

These findings suggest that reducing catastrophic failures (ruptures) may have greater practical impact on user retention than equivalent improvements in mean performance.

**Connection to Theoretical Guarantees.**    The rupture reduction can be understood through Theorem 2.2: triangular fixed-point refinement contracts the representation space with factor $\rho < 1$ per iteration. After $K$ iterations, worst-case deviation is bounded by $\rho^K \cdot \epsilon_0$, where $\epsilon_0$ is the initial cross-space disagreement. This geometric contraction specifically targets tail events—large initial deviations are reduced exponentially, converting potential ruptures into bounded inconsistencies.

## Large Language Model Usage Statement

We used large language models (GPT-5.2 and Claude Opus 4.5) solely as auxiliary tools for (i) manuscript language polishing (grammar, phrasing, and clarity) and (ii) organizing and summarizing relevant literature during the related-work preparation. All technical contributions, methodological designs, experimental procedures, and scientific conclusions were developed and verified by the authors, who take full responsibility for the content of this paper.

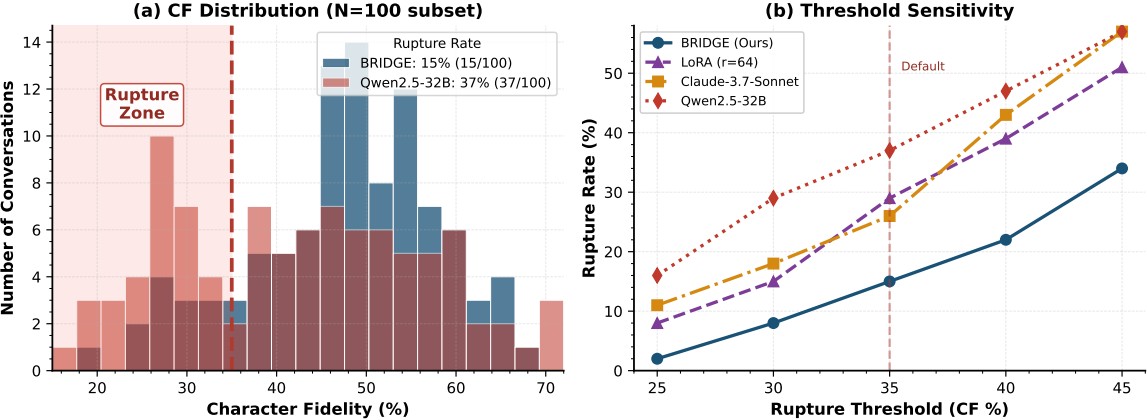

*Figure 8.* **Rupture risk analysis.** (a) Character Fidelity distribution showing rupture zone (CF $< 35\%$). BRIDGE shifts mass away from the tail compared to Qwen2.5-32B. (b) Threshold sensitivity: relative method ordering is stable across $\tau \in [25\%, 45\%]$.

