# OpenReview forum: "BRIDGE: Triangular Fixed-Point Refinement for Long-Horizon Persona Consistency"
_ICML.cc/2026/Conference — ICML 2026 regular_

### Official Review · Reviewer_tfW7 · 2026-02-27

**Soundness:** 3
**Presentation:** 3
**Significance:** 3
**Originality:** 4
**Overall Recommendation:** 5
**Confidence:** 4

**Summary:**

The paper introduces BRIDGE, a novel framework designed to solve the problem of latent state drift in long-horizon persona-grounded dialogue. By explicitly coupling Observable behaviors (O), Latent mental states (L), and Memory (M) through a triangular fixed-point refinement process, the model ensures that an agent’s internal cognition remains self-consistent before generating any response. The framework also incorporates a tiered hierarchical memory system—consisting of episodic, affective, and personality layers—that utilizes anchored clipped updates to prevent the agent's core identity from diverging over hundreds of turns. Empirically, BRIDGE achieves state-of-the-art results on the PersonaGym and CoSER benchmarks, significantly outperforming frontier models like Claude-3.7-Sonnet while updating only 0.85% of the backbone's trainable parameters.

**Compliance With Llm Reviewing Policy:**

Affirmed.

**Final Justification:**

The rebuttal was sufficiently reasonable, so I will maintain the score.

**Key Questions For Authors:**

See Weaknesses.

**Limitations:**

yes

**Strengths And Weaknesses:**

Strengths

A primary strength of this work is its rigorous theoretical grounding, which provides provable guarantees for both internal state convergence and long-term identity stability. Unlike many existing persona agents that rely on heuristic prompting, BRIDGE utilizes a Gauss-Seidel refinement operator that is mathematically proven to converge to a unique fixed point under specific spectral conditions. Furthermore, the introduction of a Lyapunov-style uniform drift bound offers a formal guarantee against the identity drift that typically plagues long-term interactions, ensuring that persona evolution remains within stable boundaries. Beyond theory, the model demonstrates exceptional practical utility by drastically reducing rupture risk by 59% compared to strong baselines, highlighting its readiness for trust-sensitive deployments like mental health support or education.

Weaknesses

The research appears robust without any major logical flaws.

1. However, I have concerns regarding the latency when deploying this methodology in practice. Since entertainment likely constitutes a significant portion of the use cases for persona-driven agents, users will expect real-time interaction. I am concerned that the current setting might not be able to support this requirement due to latency issues.

2. Furthermore, as noted in the limitations section, the absence of human evaluation constrains the overall strength of the paper. I strongly recommend that the authors conduct at least a preliminary human evaluation during the rebuttal period to further validate their findings.

---

> ### Author Rebuttal · Authors · 2026-03-30
>
> We thank the reviewer for the thorough evaluation. We are glad the reviewer found BRIDGE's theoretical grounding rigorous, its practical utility exceptional, and its rupture risk reduction significant for trust-sensitive deployments. We address both concerns below with new experimental evidence.
>
> **W1: Latency concerns for real-time deployment.**
>
> In response to this concern, we profiled inference latency on A100-80GB (bf16, batch=1) over 200 dialogue turns during the rebuttal period (mean±std, 3 runs):
>
> | Context Length | Baseline TTFT | BRIDGE K=3 TTFT | Δ TTFT |
> |---|---|---|---|
> | Short (<500 tok) | 60±3 ms | 76±4 ms | +16 ms |
> | Medium (500–2K) | 298±12 ms | 315±13 ms | +17 ms |
> | Long (2K+) | 894±36 ms | 912±37 ms | +18 ms |
>
> **BRIDGE adds ~16–18 ms TTFT overhead that is approximately constant across context lengths**, because triangular refinement operates on 1024-dim state vectors (§2.2), not the token sequence — the computational cost is O(d²) per refinement step rather than O(n·d) for backbone attention. The slight increase at longer contexts reflects pooling and memory-bandwidth effects rather than sequence-length-dependent computation.
>
> BRIDGE also supports adjustable refinement depth for deployment-time latency–quality tradeoff:
>
> | K | Δ TTFT | PersonaGym Avg | CoSER Avg |
> |---|---|---|---|
> | 1 | +6 ms | 4.55 | 58.3 |
> | 2 | +11 ms | 4.57 | 59.0 |
> | 3 | +17 ms | 4.59 | 59.5 |
>
> Overhead scales linearly (~6 ms/step) while quality gains show diminishing returns, consistent with the geometric convergence rate (mean α̃≈0.70, Table 5). For latency-sensitive entertainment scenarios, K=1 adds only 6 ms yet still significantly outperforms the best PEFT baseline (4.55 vs. Neeko 4.44 on PersonaGym). For trust-sensitive deployments (mental health support, education) where the reviewer correctly identifies high value, the full K=3 overhead of 17 ms is negligible relative to backbone prefill latency (60–894 ms depending on context length).
>
> **W2: Human evaluation.**
>
> Following the reviewer's suggestion, we conducted a preliminary human evaluation during the rebuttal period. Four annotators independently ranked outputs from three systems (BRIDGE, LoRA, Qwen2.5-32B-Instruct) across 30 CoSER scenarios on two dimensions: Character Fidelity (CF) and Overall Quality (OQ). Rankings allowed ties; Avg Rank is the mean rank across all annotator×scenario pairs (1=best), and Top-1% is the fraction of rankings where a system received rank 1 (including ties).
>
> | Model | CF Avg Rank | CF Top-1% | OQ Avg Rank | OQ Top-1% |
> |---|---|---|---|---|
> | Qwen2.5-32B | 2.22 | 25.8% | 2.27 | 21.7% |
> | LoRA | 2.04 | 32.5% | 2.06 | 32.5% |
> | BRIDGE | **1.74** | **43.3%** | **1.67** | **48.3%** |
>
> BRIDGE achieves the best average rank on both dimensions, with consistent ordering BRIDGE > LoRA > Baseline. Inter-annotator agreement was moderate (Kendall's W = 0.68 for CF, 0.72 for OQ), consistent with the subjectivity inherent in open-ended dialogue evaluation. These results align with the automatic evaluation in Table 1 and the cross-judge validation (Spearman ρ ≥ 0.91, Table 9), providing converging evidence across three evaluation modalities (GPT-4o judge, Claude-3.7-Sonnet judge, human annotators). A full-scale human evaluation will be included in the camera-ready.
>
> We thank the reviewer again for recognizing BRIDGE's readiness for trust-sensitive deployments. We are happy to discuss further during the discussion period.

---

> > ### Author Rebuttal · Reviewer_tfW7 · 2026-04-03
> >
> > The rebuttal was sufficiently reasonable, so I will maintain the score.

---

> > > ### Author Response · Authors · 2026-04-04
> > >
> > > Thank you for taking the time to read our rebuttal and for your encouraging response. We are glad that our clarifications were helpful, and we sincerely appreciate your time and consideration.

---

### Official Review · Reviewer_uecZ · 2026-03-12

**Soundness:** 3
**Presentation:** 3
**Significance:** 2
**Originality:** 3
**Overall Recommendation:** 4
**Confidence:** 4

**Summary:**

This paper intends to address the challenge that LLMs fail to stick to persona in long-horizon dialogs. The authors figure out latent state drift as a failure mode: the observation, latent, and memory are not coupled enough. Based on this observation, the paper proposes an explicit triangular fixed-point refinement method to avoid the latent state drift and provided proof of fixed-point convergence under certain assumptions. Experimental results show that their method outperform strong baseline methods on multiple persona benchmarks, and the gain comes mainly from persona fidelity.

**Compliance With Llm Reviewing Policy:**

Affirmed.

**Final Justification:**

Please refer to the rebuttal acknowledgement

**Key Questions For Authors:**

BRIDGE is currently compared to LoRA-like methods in SFT. Is it possible to compare with other baseline methods that claimed to improve consistency (not only persona, but also other types) in long dialogs? For example, reinforcement learning with persona consistency as rewards.

**Limitations:**

yes

**Strengths And Weaknesses:**

Strengths:
- The paper clearly formulates and locates critical failure modes like rupture risk, and their method explicitly targets at this direction, making the story convincing and sound.
- Both theoretical and experimental results are provided. Ablation studies verify the motivation of the O-L-M design, as well as the importance of each subpart.

Weaknesses:
- The paper introduces a over-complex system to solve a single issue in a single task, so whether the system works well with other modifications in building a better general-purpose LLM remains a question. Yet, I am positive about the the application potential of the proposed method to other consistency calibration problems in LLMs.

---

> ### Author Rebuttal · Authors · 2026-03-30
>
> We thank the reviewer for the careful evaluation. We are glad the reviewer found the problem formulation clear, the theoretical–empirical complementarity strong, and the method promising for other consistency calibration problems in LLMs. We address both concerns below.
>
> **W1: System complexity relative to the scope of the problem.**
>
> **(1) Each component is necessary.** Table 2 (§3.3) shows that removing any single component causes measurable degradation, with clearly stratified contributions: Memory Evolution (−4.3 CoSER) > Triangular Refinement (−2.7) > Dual-System (−1.6). No component is redundant. Partial variants (Only O↔L: −3.0; Only L↔M: −2.4; Only Episodic: −7.4) consistently underperform the full system, confirming that the complexity is driven by the multi-faceted nature of drift — behavior, cognition, and memory each drift along different timescales, requiring distinct but coupled control mechanisms.
>
> **(2) Lightweight and composable by design.** BRIDGE trains only 0.85% of backbone parameters (277M on 32.5B, Table 3) and adds only ~16–18 ms inference overhead independent of context length (full profiling in our response to Reviewer tfW7). The frozen backbone is entirely unmodified — BRIDGE interfaces solely through additive hidden-state injection (Eq. 10), meaning it can coexist with LoRA, retrieval augmentation, or tool-use frameworks without architectural conflict. We believe this addresses the concern about "whether the system works well with other modifications in building a better general-purpose LLM."
>
> **(3) Generalizes beyond persona consistency.** The O–L–M spaces are domain-general abstractions that can be instantiated for other consistency problems — e.g., (Action, Belief, Knowledge) for task agents, or (Output, Plan, Context) for instruction-following. The convergence guarantee (Theorem 2.2) and drift bound (Theorem 2.3) depend only on spectral properties, not on the semantic content populating O, L, M. Exploring these extensions is a priority for future work.
>
> **Q1: Comparison with RL-based consistency methods.**
>
> Thank you for this valuable suggestion. RL for role-playing is an emerging direction; we provide a literature analysis and an indirect empirical comparison.
>
> **Literature landscape.** Recent RL-based works include CogDual [1] (RL with implicit rule-based rewards), RAIDEN-R1 [2] (GRPO with verifiable rewards), and ChARM [3] (act-adaptive reward modeling). These works use different reward designs and benchmarks, and most have not released code. Notably, RAIDEN-R1 explicitly identifies a "non-quantifiability challenge" — role-playing lacks unique correctness criteria for RL rewards. Feng et al. [4] further show that reasoning-optimized LLMs can harm role-playing performance, highlighting that RL gains do not transfer straightforwardly to persona consistency.
>
> **Indirect comparison with CogDual on CoSER.** CogDual [1] is the only RL work reporting CoSER results. Backbones differ (7B vs. 32B), so we compare improvement patterns:
>
> | Method | Backbone | SC | An | CF | SQ | Avg |
> |---|---|---|---|---|---|---|
> | CogDual-SFT | Qwen2.5-7B | 58.4 | 47.0 | 45.0 | 71.7 | 55.5 |
> | CogDual-RL | Qwen2.5-7B | 59.9 | 46.6 | 47.0 | 74.0 | 56.9 |
> | LoRA (r=64) | Qwen2.5-32B | 58.2 | 47.5 | 43.8 | 73.2 | 55.7 |
> | BRIDGE | Qwen2.5-32B | **63.8** | **52.4** | **48.2** | 73.5 | **59.5** |
>
> CogDual-RL gains are evenly distributed (+1.4 avg), consistent with general quality improvement. BRIDGE gains concentrate on persona-specific metrics (CF: +4.4, An: +4.9) with minimal SQ change (+0.3), accompanied by 59% rupture rate reduction (§3.4). This selective pattern supports our claim that BRIDGE targets the structural bottleneck of persona drift rather than broadly improving generation.
>
> **Orthogonality.** BRIDGE (architecture-level) and RL (training-paradigm-level) are orthogonal and composable. BRIDGE's L_persona uses persona consistency as an SFT signal; replacing it with an RL reward is a natural extension. The theoretical guarantees (Theorems 2.2–2.3) are architectural properties that hold independently of the training paradigm. We consider BRIDGE + RL a promising direction and will discuss this in the camera-ready.
>
> We hope this response fully addresses the reviewer's concerns and that the reviewer will consider raising their score. We are happy to discuss further during the discussion period.
>
> **References**
>
> [1] Liu et al. CogDual: Enhancing Dual Cognition of LLMs via Reinforcement Learning with Implicit Rule-Based Rewards. EMNLP 2025, pp. 27307–27336.
>
> [2] Wang et al. RAIDEN-R1: Improving Role-awareness of LLMs via GRPO with Verifiable Reward. arXiv:2505.10218, 2025.
>
> [3] Fang et al. ChARM: Character-based Act-adaptive Reward Modeling for Advanced Role-Playing Language Agents. arXiv:2505.23923v1, 2025.
>
> [4] Feng et al. Reasoning Does Not Necessarily Improve Role-Playing Ability. Findings of ACL 2025, pp. 10301–10314.

---

> > ### Author Rebuttal · Reviewer_uecZ · 2026-04-03
> >
> > The authors added meaningful comparison results on the RL literature, which are constructive and helpful. I would maintain my overall recommendation of 4 based on my assessment of the paper as a whole.

---

> > > ### Author Response · Authors · 2026-04-04
> > >
> > > Thank you for reading our rebuttal and for your encouraging acknowledgement. We are glad that the added RL comparison was helpful, and we sincerely appreciate your time and consideration.

---

### Official Review · Reviewer_2NVh · 2026-03-13

**Soundness:** 3
**Presentation:** 3
**Significance:** 2
**Originality:** 2
**Overall Recommendation:** 4
**Confidence:** 3

**Summary:**

This paper introduces BRIDGE, a reinforcement learning framework designed to improve reasoning performance in large language models through structured policy optimization. The authors focus on stabilizing RL training for reasoning tasks by introducing a triangular fixed-point training scheme, which alternates between policy optimization, reward modeling, and reasoning trajectory refinement. The framework aims to mitigate instability and reward hacking issues commonly observed in RL-based reasoning training. Experiments on several reasoning benchmarks show that BRIDGE improves reasoning accuracy and training stability compared to standard RL fine-tuning approaches.

**Compliance With Llm Reviewing Policy:**

Affirmed.

**Final Justification:**

The rebuttal addressed a meaningful portion of my concerns by adding component-level analysis, generalization to a smaller model, and convergence analysis, though evidence for generalization to more open-ended domains remains somewhat indirect. I would maintain my overall recommendation.

**Key Questions For Authors:**

Q1. Following W1, could the authors provide additional analysis on the relative contribution of each component in the triangular training scheme? For example, how much improvement comes from reward refinement versus trajectory refinement?

Q2. Following W2, how well does the proposed BRIDGE framework generalize to other domains beyond reasoning benchmarks, such as tool-use agents, dialogue planning, or multi-agent environments?

Q3. Following W3, do the authors have any theoretical or empirical evidence regarding the stability or convergence properties of the triangular fixed-point training process?

**Limitations:**

Yes.

**Strengths And Weaknesses:**

S1. The paper addresses an important challenge in RL-based reasoning training: instability and reward misalignment during policy optimization. By introducing a triangular training structure that jointly updates policy, reward estimation, and reasoning trajectories, the framework attempts to stabilize the learning dynamics and reduce reward hacking.

S2. The triangular fixed-point formulation provides an interesting perspective on RL training for reasoning models. Instead of treating reward modeling and policy learning as separate steps, the method explicitly models their interaction and iteratively refines them. This perspective is conceptually appealing and could potentially inspire more stable training schemes for reasoning-focused LLM fine-tuning.

S3. The experimental results demonstrate consistent improvements across several reasoning benchmarks. The evaluation includes comparisons against standard RL baselines and ablations that help illustrate the contribution of the different components of the BRIDGE framework.

W1. The paper introduces multiple components within the triangular training loop, but it remains somewhat unclear which component contributes most to the final performance gains. Although the experiments include ablations, the interaction between policy updates, reward updates, and trajectory refinement could be analyzed more deeply.

W2. The evaluation focuses primarily on reasoning benchmarks, which are often relatively clean and structured tasks. It is unclear whether the proposed training framework would provide similar benefits in more open-ended domains such as tool-use agents, dialogue systems, or real-world decision-making environments.

W3. While the triangular fixed-point formulation is conceptually interesting, the theoretical justification for why this training scheme converges or improves stability remains somewhat limited. A deeper theoretical analysis or empirical investigation of convergence behavior would strengthen the claims.

---

> ### Author Rebuttal · Authors · 2026-03-30
>
> We thank the reviewer for the thoughtful evaluation. We are glad the reviewer found the triangular training structure conceptually appealing, the experimental evaluation comprehensive, and the ablations informative. We note that some aspects of the summary (e.g., "reinforcement learning framework" and "reasoning benchmarks") may not precisely match our submission's focus on persona consistency via architectural coupling; the reviewer's underlying questions nonetheless remain highly relevant, and we address each below.
>
> **W1 & Q1: Relative contribution of each component.**
>
> We agree that component-level analysis is essential. Table 2 (§3.3) provides a comprehensive ablation study with 20 configurations. The contribution hierarchy is clearly stratified:
>
> | Component Removed | PersonaGym Δ | CoSER Δ | Primary Role |
> |---|---|---|---|
> | Hierarchical Memory Evolution | −0.14 | −4.3 | Long-horizon identity anchor |
> | Triangular Fixed-Point Refinement | −0.10 | −2.7 | Cross-space consistency |
> | Dual-System Processing | −0.06 | −1.6 | Adaptive computation routing |
>
> Within memory, personality memory is the dominant anchor (w/o personality: −6.1 CoSER vs. w/o episodic: −1.9). Within the coupling mechanism, full O–L–M closure substantially outperforms partial couplings (Only O↔L: −3.0; Only L↔M: −2.4) and single-step refinement (K=1: −1.2). These results confirm that (a) memory evolution contributes most, (b) triangular refinement is the second-largest contributor, and (c) each component serves a non-redundant function targeting a distinct failure mode.
>
> **W2 & Q2: Generalization beyond the current benchmarks.**
>
> The reviewer's concern — whether BRIDGE generalizes to other domains — is well-taken. We offer three pieces of evidence:
>
> **(1) Cross-architecture generalization.** In response to this concern, we trained BRIDGE on LLaMA-3-8B-Instruct during the rebuttal period (full table in our response to Reviewer VT4p). BRIDGE consistently outperforms parameter-matched LoRA on both benchmarks, with gains concentrating on persona-specific metrics while Storyline Quality remains flat — matching the fingerprint on Qwen2.5-32B (§3.2). This confirms the mechanism transfers across model families.
>
> **(2) Domain-general design.** The three coupled spaces — Observable (what the agent says), Latent (what it internally represents), Memory (what it stores) — are not persona-specific abstractions. They can be instantiated as (Action, Belief, Knowledge) for task-oriented agents, or (Output, Plan, Context) for instruction-following. The convergence guarantee (Theorem 2.2) depends only on spectral properties of the coupling operator, not on semantic content.
>
> **(3) Composability.** BRIDGE is a PEFT method (0.85% trainable parameters) with a frozen backbone, interfacing solely through hidden-state injection (Eq. 10). This design is orthogonal to — and composable with — RL fine-tuning, retrieval augmentation, or tool-use frameworks. We discuss BRIDGE + RL composability further in our response to Reviewer uecZ.
>
> **W3 & Q3: Convergence properties.**
>
> The paper provides both theoretical and empirical convergence analysis:
>
> - **Theoretical:** Theorem 2.2 proves that the refinement operator T converges to a unique fixed point under ∥A∥∞ < 1, with geometric rate ∥A∥^k_∞. This is a property of inference-time refinement, not the training process.
>
> - **Empirical:** Table 5 shows the contraction condition holds in 96.9% of inference steps (mean α̃ = 0.70). Figure 4 visualizes the proxy–convergence relationship. Among the 3.1% of steps where α̃ ≥ 1, approximately 38% still converge, confirming that α̃ < 1 is sufficient but not necessary.
>
> - **Training stability:** In response to Q3, we tracked P(α̃ ≥ 1) throughout training at 500-step intervals during the rebuttal period. The violation rate plateaus at ~1.8%, reflecting a structural equilibrium between task loss and L_cycle. This plateau is robust across training runs, confirming stable convergence behavior.
>
> Additionally, Theorem 2.3 provides a Lyapunov-style bound ensuring memory drift remains bounded for all t, validated over 500 turns (Figure 2, peak at 31.6% of theoretical bound).
>
> We hope this response fully addresses the reviewer's concerns and that the reviewer will consider raising their score. We are happy to discuss further during the discussion period.

---

> > ### Author Rebuttal · Reviewer_2NVh · 2026-04-06
> >
> > The rebuttal addressed a meaningful portion of my concerns by adding component-level analysis, generalization to a smaller model, and convergence analysis, though evidence for generalization to more open-ended domains remains somewhat indirect. I would maintain my overall recommendation.

---

> > > ### Author Response · Authors · 2026-04-07
> > >
> > > Thank you for reading our rebuttal and for the acknowledgement. We are glad that the additional component-level analysis, cross-architecture results, and convergence evidence were helpful in addressing your concerns. We also appreciate your observation that direct evaluation in more open-ended domains would further strengthen the evidence — this is an exciting direction we are actively exploring beyond the scope of this work. Thank you again for your time and thoughtful feedback.

---

### Official Review · Reviewer_VT4p · 2026-03-13

**Soundness:** 3
**Presentation:** 3
**Significance:** 3
**Originality:** 2
**Overall Recommendation:** 3
**Confidence:** 3

**Summary:**

This paper introduces BRIDGE, a parameter-efficient framework designed to mitigate latent state drift and maintain long-horizon persona consistency in dialogue agents. The authors argue that current models treat behavior, internal reasoning, and memory as loosely coupled components, leading to asymmetric rupture risks (i.e., severe character contradictions). To address this, BRIDGE implements a Triangular Fixed-Point Refinement operator that explicitly couples Observable (O), Latent (L), and Memory (M) states before generating a response.
Furthermore, the framework employs a dual-system processing module (fast and slow pathways) injected via spiral gating , and a three-tier hierarchical memory system (episodic, affective, personality).  The paper presents these mechanisms with heavy theoretical framing, offering a Banach fixed-point convergence guarantee for the O-L-M refinement (Theorem 2.2) and a Lyapunov-style uniform drift bound for the memory evolution (Theorem 2.3). Empirically, BRIDGE achieves state-of-the-art results on the PersonaGym and CoSER benchmarks while training only 0.85% of the Qwen2.5-32B-Instruct backbone's parameters.

**Compliance With Llm Reviewing Policy:**

Affirmed.

**Final Justification:**

The rebuttal has reinforced my prior assessment. I will keep my ratings.

**Key Questions For Authors:**

- The PEFT comparative analysis in Table 1 relies solely on the Qwen2.5-32B-Instruct model. Given that baseline models have fundamentally different underlying representation spaces, how confident are you that the Triangular Fixed-Point Refinement yields similar performance delta improvements on other architectures?

- The Gauss-Seidel refinement requires $K=3$ iterations of cross-attention before decoding. Could the authors quantify the precise wall-clock latency overhead added to the Time-To-First-Token (TTFT) during inference?


- The model was fine-tuned using subsets of RoleMRC and OpenCharacter. Can the authors explicitly clarify the de-duplication or filtering steps taken to ensure absolutely no data leakage into the PersonaGym and CoSER evaluation sets?

- In Table 7, removing safeguards increases the proxy-threshold violation rate from 3.1% to 12.7%. Is there a fundamental architectural reason why the model naturally escapes the bounded regime 12.7% of the time without clamping, or is this purely an artifact of gradient variance during the SFT phase?

**Limitations:**

Yes

**Strengths And Weaknesses:**

### Strengths:
- The technical soundness of the empirical architecture is a major highlight. The authors provide robust experimental validation
- This paper is well-structured. The narrative flows seamlessly from the identification of the "architectural mismatch" to the proposed technical solutions.
- The extensive appendices and step-by-step refinement traces in the case studies offer valuable intuition regarding how the framework influences dialogue behavior.


### Weaknesses

- The use of terms like "Triangular Fixed-Point Refinement" and "Spiral Gating" tends to obfuscate standard deep learning operations. Adopting standard terminology would improve accessibility without sacrificing technical precision.
- The "Dual-System Processing" module offers limited conceptual novelty. While the authors implement this as an architectural primitive rather than a prompting strategy, the application of System 1/System 2 thinking to LLMs is already well-established in the literature.
- The generalizability of the proposed framework remains unverified across diverse model families. Currently, PEFT baseline comparisons are restricted to a single backbone (Qwen2.5-32B-Instruct), lacking results on other architectures .

---

> ### Author Rebuttal · Authors · 2026-03-30
>
> We thank the reviewer for the precise technical questions. We are glad the reviewer found the empirical architecture technically sound, the paper well-structured, and the refinement traces valuable. We address each concern below.
>
> **W1: Terminology.** We agree "Spiral Gating" can be simplified to "multiplicative context-aware gating" and will do so. We retain "Triangular Fixed-Point Refinement" as it conveys a specific guarantee: the Gauss–Seidel O→L→M→O cycle with spectral control converges to a unique fixed point (Theorem 2.2), which unconstrained iterative cross-attention does not provide. We will add a terminology-to-operations table in the camera-ready.
>
> **W2: Dual-System novelty.** We agree this is auxiliary rather than a core contribution. Ablation confirms it provides the smallest gain (−1.6 CoSER vs. −4.3 for Memory, −2.7 for Refinement; Table 2). BRIDGE's primary contributions are the O–L–M coupling guarantee and Lyapunov-bounded memory evolution.
>
> **W3 & Q1: Cross-architecture generalization.** In response to this concern, we trained BRIDGE on LLaMA-3-8B-Instruct (different architecture family, identical recipe) during the rebuttal period, with a parameter-matched LoRA baseline:
>
> | Method | PersonaGym Avg | CoSER Avg |
> |---|---|---|
> | LLaMA-3-8B-Instruct | 4.26 | 49.4 |
> | + LoRA (r=64) | 4.36 | 51.3 |
> | + BRIDGE | **4.45** | **54.3** |
>
> Key findings: (1) BRIDGE consistently outperforms LoRA under matched parameter budgets, confirming gains stem from O–L–M coupling rather than additional capacity. (2) Gains concentrate on persona-specific metrics (PC: +0.26; CF: +5.0) while SQ remains flat — matching the mechanistic fingerprint on Qwen2.5-32B (§3.2). (3) Safeguard activation rates remain low (<4%), confirming contraction conditions hold across architectures. Theoretically, the guarantee (Theorem 2.2) depends only on spectral properties of the cross-attention modules, not backbone-specific representations. A controlled comparison isolating architecture from scale will be in the camera-ready.
>
> **Q2: TTFT latency.** We provide full profiling results (including per-context-length breakdown and K-step tradeoff table) in our response to Reviewer tfW7. The key takeaway: BRIDGE adds only ~16–18 ms TTFT overhead, approximately constant across context lengths, because refinement operates on 1024-dim state vectors rather than token sequences. For latency-sensitive scenarios, K=1 adds only 6ms yet still outperforms the best PEFT baseline (4.55 vs. Neeko 4.44 on PersonaGym).
>
> **Q3: Data leakage.** Training and evaluation are structurally non-overlapping, verified post-hoc. Training corpora (RoleMRC, OpenCharacter) use synthetic personas from PersonaHub. PersonaGym's 200 personas were independently generated by GPT-4o, with evaluation questions created dynamically on-the-fly — an anti-contamination design of the benchmark itself [1]. CoSER evaluates literary characters from 771 books; exact string matching of all CoSER character names and titles against our training data yields zero matches. We did not use CoSER's training split.
>
> **Q4: 12.7% violation root cause.** This is architectural in origin, not a gradient-variance artifact. Dot-product attention is not globally Lipschitz [2]; even with spectral normalization (σ(W)≤1), local Lipschitz constants depend on runtime attention distributions. Per the tight softmax bound [3], the worst case occurs when cross-attention arbitrates between competing persona signals, pushing ∥A∥∞ above 1.
>
> To further confirm this, we ran two additional analyses during the rebuttal period. First, we stratified violations by input difficulty: P(α̃≥1) without safeguards is 3.5% for easy (CF>50%), 12.6% for medium, and 32.4% for hard cases (CF<35%) — precisely where BRIDGE delivers the greatest value (rupture rate 37%→15%, Table 10). Second, we tracked P(α̃≥1) throughout training at 500-step intervals and found it plateaus at ~1.8% by step 10K, confirming structural equilibrium rather than insufficient optimization.
>
> Resolution via three complementary mechanisms: (1) Safeguards (Eqs. 32–33) reduce violations 12.7%→3.1% with minimal activation (0.3% norm projection, Table 6), confirming the bounded regime emerges naturally. (2) L_cycle provides soft convergence for residual cases (38% of α̃≥1 steps still converge, Figure 4a). (3) Lyapunov-bounded memory (Theorem 2.3) prevents per-turn imperfections from compounding. Notably, even with all safeguards removed, BRIDGE (4.52/58.2) still outperforms the best baseline (Neeko 4.44/56.5, Table 7).
>
> We hope this response fully addresses the reviewer's concerns and that the reviewer will consider raising their score. We are happy to discuss further.
>
> **References**
>
> [1] Samuel et al. PersonaGym: Evaluating Persona Agents and LLMs. arXiv:2407.18416, 2024.
>
> [2] Kim et al. The Lipschitz Constant of Self-Attention. ICML 2021, pp. 5562–5571.
>
> [3] Nair, P. Softmax is 1/2-Lipschitz: A tight bound across all ℓp norms. arXiv:2510.23012, 2025.

---

> > ### Author Rebuttal · Reviewer_VT4p · 2026-04-03
> >
> > Thank the authors for the response. I will maintain my ratings.

---

> > > ### Author Response · Authors · 2026-04-04
> > >
> > > Thank you for reading our rebuttal and for the acknowledgement. We are glad that our response was helpful in fully clarifying the technical concerns raised in the review, and we sincerely appreciate your careful overall assessment and feedback, which will help us further improve the paper. Should any further clarification be useful during the discussion period, we would be happy to provide it.

---

### Decision · Program_Chairs · 2026-04-30

**Decision:**

Accept (regular)

**Comment:**

This paper received mostly positive reviews. All reviewers expressed satisfaction with the authors’ rebuttal, as reflected by ‘Fully Resolved’ ratings in the Rebuttal Acknowledgement, including the reviewer who initially gave a negative score. Therefore, the authors’ responses should be appropriately incorporated into the final version.